# TIT-Score: Evaluating Long-Prompt Based Text-to-Image Alignment via Text-to-Image-to-Text Consistency

## Abstract

With the rapid advancement of large multimodal models (LMMs), recent text-to-image (T2I) models can generate high-quality images and demonstrate great alignment to short prompts. However, they still struggle to effectively understand and follow long and detailed prompts, displaying inconsistent generation. To address this challenge, we introduce **LPG-Bench**, a comprehensive benchmark for evaluating long-prompt-based text-to-image generation. LPG-Bench features 200 meticulously crafted prompts with an average length of over 250 words, approaching the input capacity of several leading commercial models. Using these prompts, we generate 2,600 images from 13 state-of-the-art models and further perform comprehensive human-ranked annotations. Based on LPG-Bench, we observe that state-of-the-art T2I alignment evaluation metrics exhibit poor consistency with human preferences on long-prompt-based image generation. To address the gap, we introduce a novel **zero-shot** metric based on text-to-image-to-text consistency, termed **TIT**, for evaluating long-prompt-generated images. The core concept of TIT is to quantify T2I alignment by directly comparing the consistency between the raw prompt and the LMM-produced description on the generated image, which includes an efficient score-based instantiation **TIT-Score** and a large-language-model (LLM) based instantiation **TIT-Score-LLM**. Extensive experiments demonstrate that our framework achieves superior alignment with human judgment compared to CLIP-score, LMM-score, *etc.*, with TIT-Score-LLM attaining a **7.31%** absolute improvement in pairwise accuracy over the strongest baseline. LPG-Bench and TIT methods together offer a deeper perspective to benchmark and foster the development of T2I models. All resources will be made publicly available.

## 1 Introduction

Recent advances in large multimodal models (LMMs) have significantly improved text-to-image (T2I) generation (Zhang & Tang, 2024). State-of-the-art T2I models are now capable of producing visually compelling images and achieving strong alignment with short textual prompts (Wang et al., 2025b; Li et al., 2024). However, despite these impressive achievements, generating faithful images from long and complex prompts remains a fundamental challenge, as it requires a nuanced understanding of complex instructions and subtle visual-text alignment. While the long-prompt-based image generation is important for applications including creation, entertainment, *etc.*, existing benchmarks (Wang et al., 2025b; Li et al., 2024) mainly focus on short prompts, on which most state-of-the-art models (openai, 2025b; Google, 2025; Wu et al., 2025a) already perform well. This hides their deeper limitations in understanding long, detailed, and narrative-style instructions (Huang et al., 2025), which hinders the progress of T2I models in more complex text understanding and content creation.

To address this evaluation gap, we introduce **LPG-Bench**, a challenging new benchmark specifically designed to evaluate long-prompt T2I adherence (Hessel et al., 2021; Li et al., 2023). LPG-Bench comprises 200 meticulously designed and manually refined prompts with an average length exceeding 250 words, approaching the input capacity limits of several leading commercial models (Gao et al., 2025; Team, 2025). Based on these prompts, we generate 2,600 images using 13 state-of-

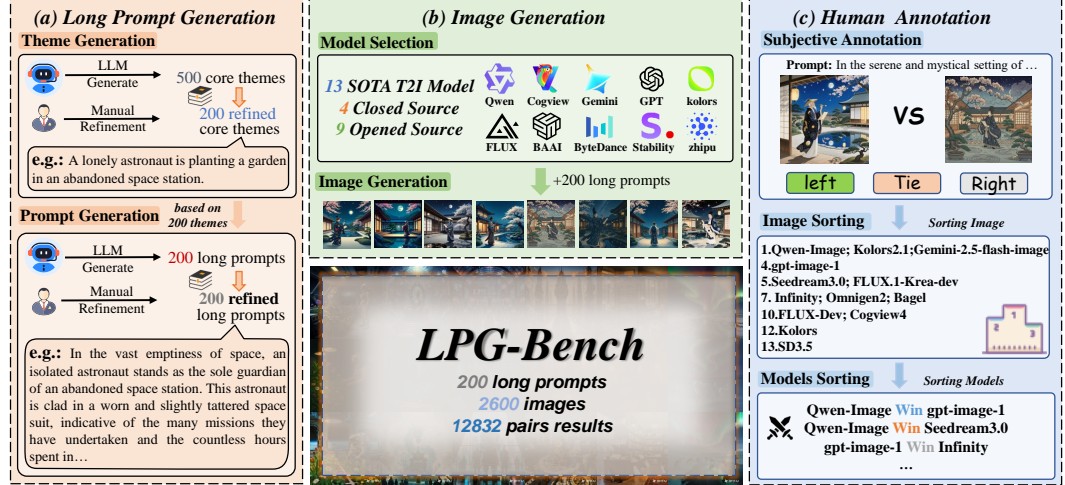

Figure 1: The construction workflow of LPG-Bench. (a) Long Prompt Generation. 200 core themes are expanded by Gemini 2.5 Pro (google, 2025) into detailed prompts averaging over 250 words. (b) Image Generation. A suite of text-to-image models produces 2,600 images from these prompts. (c) Human Annotation. The outputs are evaluated through pairwise comparisons, a process that yields 12,832 valid (non-tie) paired results for model ranking.

the-art closed-source (openai, 2025b; Google, 2025; Gao et al., 2025; Team, 2025) and open-source models (Wu et al., 2025a; Lee et al., 2025), and collect comprehensive human-ranked annotations to provide a robust foundation for evaluation.

Besides the lack of benchmarks, current T2I evaluation metrics are also insufficient in assessing the images produced using long prompts. Traditional embedding-similarity-based metrics, such as CLIP-Score (Hessel et al., 2021) suffer from information loss when compressing complex text into embeddings. Recently, some methods, such as VQA-Score (Li et al., 2024) have investigated to ask LMMs with questions for each attribute in the prompt and accumulate the answers to get a score, which is unrealistic for long prompts. Some works have explored to directly ask LMMs to give the alignment score between generated images and their prompts (Wang et al., 2025b; Huang et al., 2024), which generally lack stable, calibrated scoring framework. Some other works have studied training networks based on CLIP or LMMs (Xu et al., 2023; Wu et al., 2023; Wang et al., 2025b), however, these models require large-scale training samples and may not generalize well on long-prompt image generation.

To address these challenges, we propose a novel, **zero-shot** evaluation framework for measuring prompt-image alignment in long-prompt generation scenarios through text-to-image-to-text consistency, termed **TIT**, of which the motivation comes from the superior judgment ability of LLM for text similarity measurement (Gu et al., 2025). Our framework innovatively breaks down the complex cross-modal evaluation into two distinct steps. First, it leverages the powerful perceptual capabilities of a vision-language-models (VLMs) (Zhang et al., 2024a) to objectively translate image content into a rich textual description. Then, it calculates the semantic alignment between the original prompt and this description purely within the textual domain. Leveraging different strategies for text-consistency evaluation, we present two complementary instantiations. (1) **TIT-Score**, employs a state-of-the-art text embedding model (Zhang et al., 2025) to produce embedding vectors from texts and calculate their cosine similarity, serving as a computationally efficient and robust evaluation tool. (2) **TIT-Score-LLM**, employs a frontier large language model (LLM), such as Gemini 2.5 Pro, for the final similarity judgment.

Our comprehensive evaluation on LPG-Bench not only confirms the limitations of existing metrics (Hessel et al., 2021; Li et al., 2024; Wang et al., 2025b) on long-prompt tasks but also demonstrates the superiority of our framework. The results show a stronger alignment with human judgment, with **TIT-Score-LLM** in particular establishing a new state of the art by achieving a **7.31%** absolute improvement in pairwise accuracy over the strongest baseline (Wang et al., 2025b). Meanwhile, the standard **TIT-Score** delivers near-SOTA performance with remarkable efficiency and accessibility.

The main contributions of this paper are as follows:

- We construct and release **LPG-Bench**, the first comprehensive benchmark focused on evaluating the long-prompt adherence of T2I models.

- We propose a novel, **zero-shot** evaluation framework with two instantiations, **TIT-Score** and **TIT-Score-LLM**, which align better with human preferences on long-prompt evaluation without any training, effectively improving evaluation accuracy and reliability.

- Together, our work provides a robust foundation for benchmarking and advancing T2I models in long-text understanding, enabling a clear assessment for the core limitations of current state-of-the-art models in this critical capability.

## 2 RELATED WORKS

### 2.1 TEXT-TO-IMAGE MODELS

The text-to-image field is primarily driven by two paradigms: Diffusion (Ho et al., 2020) and Autoregressive (Tian et al., 2025) models. The diffusion paradigm has evolved from classic U-Net backbones, like in Kolors (Kolors Team, 2024), to more powerful Transformer architectures. This includes the Multimodal Diffusion Transformer (MMDiT) (Esser et al., 2024) used by Qwen-Image (Wu et al., 2025a) and Stable Diffusion 3.5 (Stability-AI, 2025), as well as flow-based variants explored in FLUX.1-Dev Labs (2024) and Bagel. The autoregressive paradigm, inspired by large language models, includes approaches like CogView4 (zai org, 2023) which combine a Transformer with a VQ-VAE (van den Oord et al., 2017), and innovations such as Infinity's bitwise modeling to enhance detail fidelity. A significant recent trend is the rise of unified multimodal models like Gemini 2.5 Flash Image (Google, 2025), gpt-image-1 (openai, 2025b), and Omnigen2 (Wu et al., 2025b), which integrate understanding and generation within a single framework, signaling a shift towards more general-purpose systems.

### 2.2 EVALUATION BENCHMARKS

The evaluation paradigm for text-to-image models has evolved significantly, moving beyond early metrics like FID (Jayasumana et al., 2024) and CLIPScore (Hessel et al., 2021) toward more sophisticated, multi-dimensional frameworks. A new generation of benchmarks aims to measure advanced capabilities across different axes of complexity. For instance, benchmarks such as T2I-CompBench++ (Huang et al., 2025) define complexity through compositionality, assessing a model's ability to combine multiple distinct objects and concepts, while others like the WISE benchmark (Niu et al., 2025) evaluate the integration of world knowledge. In a parallel track, large-scale "Arena" platforms like LMArena (Berkeley, 2025) have emerged to capture human preferences for aesthetic quality. Although these benchmarks have greatly advanced the field, their definition of 'complexity' primarily revolves around the combination of discrete, often short, conceptual elements. This leaves a crucial gap: a dedicated, standardized benchmark designed to rigorously measure a model's adherence to a single, continuous, and exceptionally long descriptive prompt.

### 2.3 EVALUATION METHODS

Automated alignment evaluation began with CLIPScore (Hessel et al., 2021), which uses embedding similarity but struggles with complex prompts. To improve accuracy, subsequent methods employed stronger backbones like BLIP-2 (Li et al., 2023) or trained reward models like PickScore (Kirstain et al., 2023) and HPSv2 (Wu et al., 2023) on human preference data to better capture subjective alignment. Other approaches, such as VQA-Score (Li et al., 2024), verify factual details by decomposing prompts into questions. The latest advancements involve fine-tuning large multimodal models (LMMs) to act as direct evaluators, as seen in LMM4LMM (Wang et al., 2025b). However, these methods still face challenges with exceptionally long prompts; embedding-based techniques suffer from information loss via compression, while direct LMM scoring lacks a stable, calibrated framework, limiting their reliability.

Table 1: Summary of Text-to-Image Models. **Legend**: **Type**: T2I (Pure Text-to-Image), MM (Unified Multimodal); **Developer**: BFL (Black Forest Labs), VSL (VectorSpaceLab), SAI (Stability AI); **Architecture**: T (Transformer), D (Diffusion), F (Flow), AR (Autoregressive).

| Model Name | Developer | Release | Type | Arch. | Parameters |
|---|---|---|---|---|---|
| Bagel (Deng et al., 2025) | ByteDance | 2025-05 | MM | MoT + VAE | 7B |
| CogView4 (zai org, 2023) | Zhipu AI | 2025-03 | T2I | T + VQ-VAE | 6B |
| FLUX.1-Krea-dev (Lee et al., 2025) | BFL | 2025-07 | T2I | F + T | 12B |
| FLUX.1-dev (Labs, 2024) | BFL | 2024-08 | T2I | F + T | 12B |
| Gemini 2.5 Flash Image (Google, 2025) | Google | 2025-08 | MM | MoE | Not Disclosed |
| gpt-image-1 (openai, 2025b) | OpenAI | 2025-04 | MM | T | Not Disclosed |
| Infinity (Han et al., 2025) | ByteDance | 2024-12 | T2I | AR + T | 2B / 8B |
| Kolors (Kolors Team, 2024) | Kuaishou | 2024-06 | T2I | D + T | 1.2B |
| Kolors 2.1 (Team, 2025) | Kuaishou | 2025-07 | T2I | D | Not Disclosed |
| Omnigen2 (Wu et al., 2025b) | VSL | 2025-06 | MM | AR + D + T | ~7B |
| Qwen-Image (Wu et al., 2025a) | Alibaba | 2025-08 | T2I | MMDiT | 20B |
| Stable Diffusion 3.5 (Stability-AI, 2025) | SAI | 2024-10 | T2I | MMDiT | 2.5B |
| SeeDream3.0 (Gao et al., 2025) | ByteDance | 2025-04 | T2I | D | Not Disclosed |

## 3 CONSTRUCTING THE LPG-BENCH

To effectively measure and challenge the capabilities of current Text-to-Image (T2I) models (Zhang & Tang, 2024) in understanding and adhering to long, complex textual instructions, we have designed and constructed a new benchmark, LPG-Bench. The construction of LPG-Bench follows a rigorous process, encompassing the meticulous curation and generation of long-form prompts, image acquisition from a diverse set of state-of-the-art models (Deng et al., 2025; Google, 2025; openai, 2025b), and high-quality human preference annotation. This section details these three core stages. These three stages are illustrated in Figure 1.

### 3.1 LONG-PROMPT CURATION AND GENERATION

The defining characteristics of the prompts in LPG-Bench are their length and richness in detail. To systematically generate high-quality long-form texts, we employed a multi-stage pipeline. First, we utilized a Large Language Model (LLM) (google, 2025) to initially generate 500 diverse themes as candidates. These themes, covering a wide range of subjects, underwent a rigorous "manual screening process" to ensure quality and uniqueness, yielding 200 high-quality, diverse core themes. Following this selection, we employed the advanced capabilities of the Gemini 2.5 Pro (google, 2025) model to elaborate on each theme, guiding it with a detailed meta-instruction for in-depth descriptions. We explicitly required a text length of no less than 250 words, ensuring that each prompt contains sufficient information and detail (Gao et al., 2025; Team, 2025). Each expanded prompt then underwent a final manual review to polish its narrative fluency and logical consistency. Through this complete process, we obtained the 200 high-quality long-form prompts that form the testing foundation of LPG-Bench. For more on long-prompt generation, as well as specific classifications and analysis, see Appendices A.3, A.6, and A.7.

### 3.2 MODEL SELECTION AND IMAGE GENERATION

To conduct a comprehensive evaluation of the current state of T2I (Text-to-Image) models, we selected 13 representative, state-of-the-art models currently available. Our selection covers a variety of technical routes and model types, including four unified multimodal large models (Bagel (Deng et al., 2025), gpt-image-1 (openai, 2025b), Gemini 2.5 Flash Image (Google, 2025), and Omnigen2 (Wu et al., 2025b)) and nine pure text-to-image models. In terms of accessibility, the selection includes four closed-source commercial models (gpt-image-1, Gemini 2.5 Flash Image, SeeDream3.0 (Gao et al., 2025), and Kolors 2.1 (Team, 2025)), as well as numerous outstanding open-source and open-weight models based on different paradigms such as Diffusion (Ho et al., 2020), Autoregressive (Tian et al., 2025), and Flow models. The complete list of models and their information is provided in Table 1. For each of the 200 prompts in LPG-Bench, we generated images using these 13 models. Finally, we generated 2,600 images.

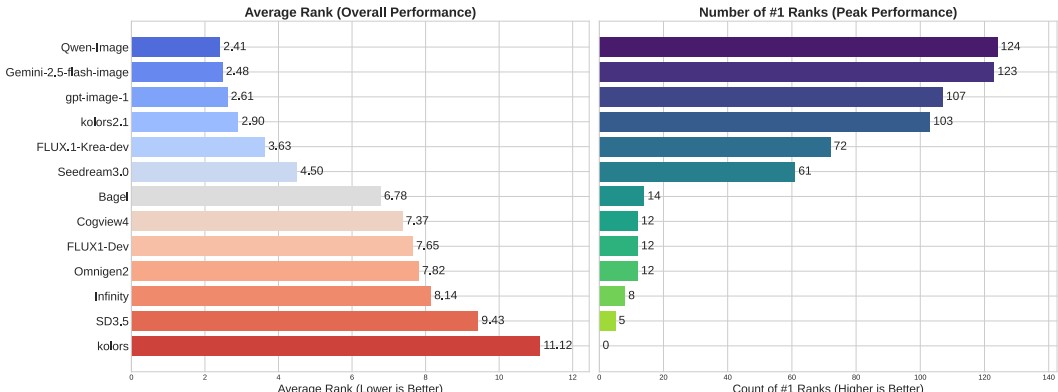

Figure 2: Model performance ranking based on human preferences. The left chart shows Average Rank (overall consistency, lower is better), while the right shows the count of #1 Ranks (peak performance, higher is better). The results reveal a clear performance hierarchy among the 13 models.

### 3.3 HUMAN PREFERENCE ANNOTATION

To address the subjectivity and poor reliability of absolute rating scales for long-prompt adherence, we designed a more reliable pairwise comparison annotation scheme. We recruited 15 annotators who, after completing a standardized training session, were asked to compare two images generated from the same prompt and select the one with superior adherence or declare a 'Tie'. The inclusion of a 'Tie' option was crucial for ensuring annotation accuracy by preventing forced choices when images were of comparable quality. To maintain high quality and consistency, a cross-review mechanism was also employed. Finally, inspired by the Merge Sort algorithm, these pairwise judgments were aggregated to compute a final relative ranking for all images corresponding to each prompt, allowing for ties. Our rigorous annotation process, specific judgment criteria, and qualitative comparison examples are detailed in Appendix A.11 and A.12.

### 3.4 ANALYSIS OF ANNOTATION RESULTS

Our human annotation results, as summarized in Figure 2, offer a detailed view of model performance on long-prompt tasks and reveal a clear hierarchy. The evaluation highlights a fierce competition at the top, with Qwen-Image (Wu et al., 2025a), Gemini-2.5-flash-image (Google, 2025), and gpt-image-1 (openai, 2025b) consistently outperforming other models. Their leading positions are confirmed by two metrics: a low average rank, which speaks to their consistent and reliable adherence to complex instructions, and a high count of #1 ranks, which demonstrates their ability to achieve peak performance and generate the most faithful images. Beyond these front-runners, a significant performance gap emerges, with many models struggling to interpret the nuanced details in the lengthy prompts. This wide spectrum of capabilities underscores that advanced textual understanding is a critical bottleneck for many T2I systems, validating LPG-Bench's role in identifying these limitations.

## 4 METHOD

Despite the abundance of existing evaluation metrics (Hessel et al., 2021; Li et al., 2023) for text-to-image generation, accurately assessing adherence to long-form prompts remains a significant challenge. Traditional automated metrics, such as CLIP-Score (Hessel et al., 2021) or BLIP-Score (Li et al., 2023), are primarily trained on image-text pairs with short descriptions. This leads to poor performance when dealing with long prompts rich in detail and complex logic, as significant information is lost during the embedding compression. On the other hand, while end-to-end scoring using modern large multimodal models (LMMs) (Zhang et al., 2024a) is an emerging trend, it suffers from inherent flaws. LMMs lack a stable, calibrated "scoring reference frame" when generating scores, resulting in inconsistent and poorly reproducible results; that is, the model itself does not precisely know the tangible difference in adherence between a score of "8" and "9".

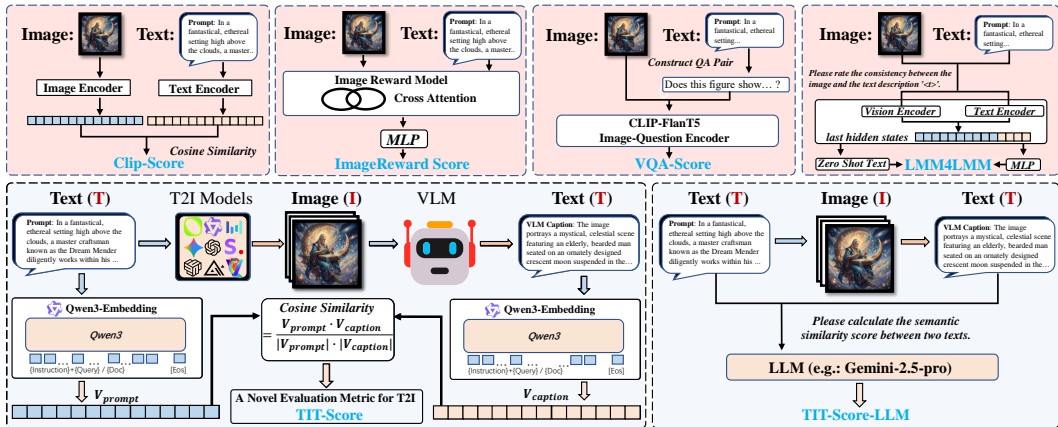

Figure 3: A comparison of Text-to-Image evaluation methods. The top row illustrates the diagrams for four baseline methods. The bottom row shows the architecture for our proposed TIT-Score and TIT-Score-LLM. TIT-Score translates an image to a text description via a VLM and then uses a text embedding model to calculate cosine similarity with the prompt; TIT-Score-LLM, however, employs a Large Language Model (LLM) for this final similarity assessment.

To address these challenges, we propose a framework built on a core design philosophy that decouples the complex text-image evaluation task. Instead of an end-to-end "black-box" scorer, we break down the evaluation into two specialized stages: (1) Visual Perception and Description, and (2) Textual Semantic Alignment. As illustrated in Figure 3, this pipeline begins with the Visual Description Generation step. Given an image $I$, we input it into a powerful vision-language model (VLM) which acts as a "describer" rather than a "judge," to objectively translate image content into a descriptive text $C_{\text{caption}}$ without any subjective scoring.

The second stage involves calculating the semantic similarity between the original prompt $P_{\text{prompt}}$ and the VLM's description $C_{\text{caption}}$, which we realize through two complementary instantiations. Our standard, embedding-based method, **TIT-Score**, uses an advanced text embedding model (Qwen3-Embedding (Zhang et al., 2025)) to encode both texts into feature vectors, $\mathbf{V}_{\text{prompt}}$ and $\mathbf{V}_{\text{caption}}$. The final score is their Cosine Similarity:

$$\text{TIT-Score} = \cos(\theta) = \frac{\mathbf{V}_{\text{prompt}} \cdot \mathbf{V}_{\text{caption}}}{\|\mathbf{V}_{\text{prompt}}\|\|\mathbf{V}_{\text{caption}}\|}$$

The score ranges from -1 to 1, with values closer to 1 indicating higher semantic consistency. Another instantiation, **TIT-Score-LLM**, also shown in Figure 3, instead prompts a powerful large language model (LLM) to directly evaluate and score the similarity between the two texts. For specifics on the VLM prompt used and other implementation details, please see Appendix A.4.

## 5 EXPERIMENTS

In this section, we conduct a series of experiments to rigorously evaluate the performance of our proposed TIT-Score. The evaluation is carried out on our LPG-Bench, leveraging the 2,600 generated images and the corresponding 12,832 non-tie human preference pairs described in Section 3.

### 5.1 EXPERIMENTAL SETUP

To provide a comprehensive assessment, we evaluate our proposed decoupled framework through its two primary instantiations: the efficient, embedding-based **TIT-Score** and the peak-performance, LLM-based **TIT-Score-LLM**. The initial visual description stage of our framework is contingent on the vision-language model (VLM) used. In our experiments, we tested a range of powerful open-source and closed-source VLMs, including models from the InternVL (Wang et al., 2025c), Qwen-VL (Bai et al., 2025), Gemma (Gemma Team et al., 2025), and MiMo-VL (Yue et al., 2025) series, as well as proprietary models like GPT-4o (openai, 2025a) and Gemini 2.5 Pro (google, 2025). We compare our methods against a diverse set of baselines representing various methodologies. These

Table 2: Main results of all metrics on the LPG-Bench benchmark. **TIT-Score** and **TIT-Score-Gemini** (This means using Gemini-2.5-pro as the LLM. ), consistently outperform all baselines. Higher values are better for Accuracy, nDCG, SRCC and KRCC. Red denotes the best value and blue denotes the second-best.

| Model / Metric | Acc (%) | SRCC | KRCC | nDCG |
|---|---|---|---|---|
| ClipScore (Hessel et al., 2021) | 48.51% | 0.0181 | 0.0154 | 0.7332 |
| BLIPv2Score (Li et al., 2023) | 53.59% | 0.0990 | 0.0775 | 0.7593 |
| HPSV2.1 (Wu et al., 2023) | 49.28% | 0.0035 | 0.0004 | 0.7333 |
| PickScore (Kirstain et al., 2023) | 49.40% | 0.0027 | 0.0014 | 0.7301 |
| MPS (Zhang et al., 2024b) | 47.80% | 0.0323 | 0.0287 | 0.7195 |
| Imagereward (Xu et al., 2023) | 50.83% | 0.0372 | 0.0259 | 0.7417 |
| VQA-Score (Li et al., 2024) | 58.20% | 0.2321 | 0.1817 | 0.7822 |
| FGA blip2 (Han et al., 2024) | 52.01% | 0.0640 | 0.0476 | 0.7364 |
| LMM4LMM (Wang et al., 2025b) | 59.20% | 0.3067 | 0.2472 | 0.7940 |
| TIT-Score (Internvl 3.5 8B) (Wang et al., 2025c) | 60.20% | 0.2496 | 0.1984 | 0.8033 |
| TIT-Score (Internvl 3.5 14B) (Wang et al., 2025c) | 61.27% | 0.2826 | 0.2183 | 0.8034 |
| TIT-Score (Internvl 3.5 30B-A3B) (Wang et al., 2025c) | 59.31% | 0.2323 | 0.1813 | 0.7960 |
| TIT-Score (Internvl 3.5 GPT-OSS) (Wang et al., 2025c) | 62.09% | 0.2988 | 0.2329 | 0.8026 |
| TIT-Score (Gemma 3n E4B) (Gemma Team et al., 2025) | 64.14% | 0.3468 | 0.2722 | 0.8101 |
| TIT-Score (Qwen2.5-VL 7B) (Bai et al., 2025) | 63.58% | 0.3280 | 0.2584 | 0.8111 |
| TIT-Score (Qwen2.5-VL 32B) (Bai et al., 2025) | 63.14% | 0.3208 | 0.2509 | 0.8106 |
| TIT-Score (Qwen2.5-VL 72B) (Bai et al., 2025) | 63.12% | 0.3189 | 0.2516 | 0.8085 |
| TIT-Score (Mimovl 7B 2508) (Yue et al., 2025) | **64.73%** | 0.3595 | **0.2810** | 0.8135 |
| TIT-Score (gpt-4o) (openai, 2025a) | 64.64% | **0.3630** | 0.2804 | **0.8209** |
| TIT-Score (Gemini 2.5 Pro) (google, 2025) | **66.38%** | **0.3953** | **0.3099** | **0.8344** |
| TIT-Score-Gemini (Gemma 3n E4B) (Gemma Team et al., 2025) | 64.97% | 0.3961 | 0.3093 | 0.8233 |
| TIT-Score-Gemini (InternVL 3.5 GPT-OSS) (Wang et al., 2025c) | 65.23% | 0.4019 | 0.3170 | 0.8285 |
| TIT-Score-Gemini (InternVL 3.5 14B) (Wang et al., 2025c) | 66.15% | 0.4114 | 0.3294 | **0.8329** |
| TIT-Score-Gemini (InternVL 3.5 8B) (Wang et al., 2025c) | 65.58% | 0.4015 | 0.3213 | 0.8311 |
| TIT-Score-Gemini (Gemini-2.5-pro) (google, 2025) | 65.58% | 0.4133 | 0.3300 | **0.8354** |
| TIT-Score-Gemini (Qwen2.5vl 7B) (Bai et al., 2025) | **66.41%** | **0.4161** | **0.3320** | 0.8279 |
| TIT-Score-Gemini (Qwen2.5vl 32B) (Bai et al., 2025) | **66.51%** | **0.4342** | **0.3416** | 0.8325 |

include traditional alignment metrics such as CLIP-Score and advanced variants like BLIPv2-Score (Li et al., 2023) and FGA-BLIP2 (Han et al., 2024). We also benchmark against a suite of reward models trained on human preferences, including PickScore (Kirstain et al., 2023), HPSv2.1 (Wu et al., 2023), ImageReward (Xu et al., 2023), and MPS (Zhang et al., 2024b). Furthermore, we include VQAScore (Li et al., 2024), a method that leverages a Visual Question Answering model to verify prompt details. To benchmark against the state of the art, we incorporate LMM4LMM (Wang et al., 2025b), an approach that fine-tunes a Large Multimodal Model (Wang et al., 2025a) for direct evaluation tasks.

To measure how well each metric's judgments align with human annotations, we employ three complementary evaluation protocols. Our primary metric is Pairwise Accuracy, which calculates the percentage of image pairs for which a metric correctly predicts the human-preferred image, computed over all 12,832 non-tie pairs from our benchmark. To evaluate the overall ranking quality for the 13 images generated for each prompt, we use ranking correlation coefficients, specifically Spearman's Rank Correlation Coefficient (SRCC) (wiki, 2025b) and Kendall's Rank Correlation Coefficient (KRCC). Furthermore, we assess the quality of the predicted rankings with a particular emphasis on the top-performing images using the Normalized Discounted Cumulative Gain (nDCG) (wiki, 2025a).

## 5.2 MAIN RESULTS

We present the main quantitative results of our comparative evaluation in this section. As shown in Table 2, the two instantiations of our decoupled framework, **TIT-Score** and **TIT-Score-LLM**, consistently and significantly outperform all selected baselines (Hessel et al., 2021; Li et al., 2024; Wang et al., 2025b; Li et al., 2023; Han et al., 2024) across all evaluation protocols. The results highlight a clear performance gap between methodologies: our **TIT-Score-LLM** achieves a peak pairwise accuracy of 66.51%, marking a 7.31% absolute improvement over the strongest baseline, LMM4LMM, and establishing the performance ceiling of our framework. Notably, our standard, embedding-based **TIT-Score** also reaches a highly competitive accuracy of 66.38%, demonstrating its value as an efficient, deployable, and near-SOTA evaluation solution.

Table 3: Comparing T2I model rankings from our framework and other evaluation models against human preference rankings.

| Model Name | Human | TIT-Score | LMM4LMM | VQA-Score | ClipScore | Blipv2Score | FGA blip2 |
|---|---|---|---|---|---|---|---|
| Qwen-Image (Wu et al., 2025a) | 1 | 1 | 1 | 1 | 1 | 1 | 2 |
| Gemini-2.5-flash-image (Google, 2025) | 2 | 3 | 8 | 3 | 9 | 12 | 3 |
| gpt-image-1 (openai, 2025b) | 3 | 2 | 3 | 2 | 4 | 5 | 8 |
| Kolors 2.1 (Team, 2025) | 4 | 5 | 4 | 12 | 13 | 9 | 11 |
| FLUX.1-Krea-dev (Lee et al., 2025) | 5 | 4 | 6 | 4 | 6 | 2 | 4 |
| SeeDream3.0 (Gao et al., 2025) | 6 | 6 | 12 | 5 | 12 | 7 | 10 |
| Bagel (Deng et al., 2025) | 7 | 7 | 2 | 7 | 8 | 3 | 7 |
| Cogview4 (zai org, 2023) | 8 | 10 | 5 | 9 | 10 | 11 | 12 |
| FLUX.1-Dev (Labs, 2024) | 9 | 12 | 9 | 13 | 11 | 6 | 9 |
| Omnigen2 (Wu et al., 2025b) | 10 | 8 | 7 | 11 | 7 | 8 | 13 |
| Infinity (Han et al., 2025) | 11 | 9 | 10 | 6 | 5 | 13 | 5 |
| SD3.5 (Stability-AI, 2025) | 12 | 11 | 11 | 8 | 2 | 4 | 1 |
| Kolors (Kolors Team, 2024) | 13 | 13 | 13 | 10 | 3 | 10 | 6 |
| SRCC wiki (2025b) | 1 | **0.929** | 0.676 | 0.626 | -0.159 | 0.302 | 0.110 |

Before delving into a detailed analysis, it is crucial to establish a threshold for statistical significance. Based on a binomial probability test for the 12,832 comparison pairs, a pairwise accuracy must exceed 50.73% to be considered statistically significant with 95% confidence ($p < 0.05$), and above 51.37% for 99.9% confidence (see Appendix A.2). Both of our proposed methods comfortably surpass this threshold, whereas many traditional metrics like ClipScore (Hessel et al., 2021) and pickscore (Kirstain et al., 2023) fail, suggesting their judgments on LPG-Bench are not reliably distinguishable from random chance.

We also report the SRCC (wiki, 2025b), KRCC, and nDCG (wiki, 2025a) scores to provide a more holistic view. However, we believe the rank correlation metrics SRCC and KRCC should be interpreted with caution in the context of LPG-Bench, primarily because the small number of items to be ranked (13) and the prevalence of ties in the ground truth can limit their stability. Nevertheless, the trends observed in these metrics generally support the conclusions drawn from pairwise accuracy, reaffirming our methods' superiority. Furthermore, analysis of Table 2 reveals a noteworthy phenomenon regarding the VLM backbone choice: a larger parameter count does not necessarily guarantee better performance. For instance, within the Qwen2.5vl (Bai et al., 2025) series, the 7B model achieved a higher accuracy than the 32B model, suggesting that raw model scale is not the sole determinant of a VLM's effectiveness in our pipeline.

To provide a more macroscopic view of ranking capabilities, we compare the average model rankings produced by various metrics against human rankings in Table 3. The leaderboard visually confirms the superiority of our framework, which achieves a high degree of correlation with human judgments (SRCC of 0.929) in overall model ranking, underscoring its reliability and advancement as a next-generation evaluation metric. A more in-depth analysis of this leaderboard is conducted in Appendix A.5.

## 5.3 ABLATION STUDY ON THE DECOUPLED DESIGN

To thoroughly validate the effectiveness of our core decoupled design and explore its different implementations, we conducted a comprehensive ablation study. As shown in Table 4, we first compared our framework against an end-to-end "LMM Score" baseline, where the VLM directly outputs a numerical score for text-image consistency. The results show that the "LMM Score" approach performs extremely poorly, with accuracies falling far below the 50% random-chance baseline (except Gemini). This strongly demonstrates the inherent instability of end-to-end scoring and validates the fundamental necessity of our "describe-then-compare" decoupled design.

We then investigated different implementations for the second stage (semantic similarity evaluation) of our framework by replacing the standard embedding model with Large Language Models (LLMs) of varying capabilities. The results reveal a clear trend: the performance of an LLM evaluator is highly and positively correlated with its own capability. Using mid-tier LMMs for self-evaluation ("Self-Eval") or the Qwen3 series (Yang et al., 2025) as evaluators yields unstable and weaker performance compared to the standard **TIT-Score**. However, when the top-tier Gemini 2.5 Pro (google, 2025) is used as the evaluator, it performs exceptionally well, even surpassing the standard **TIT-Score** in most cases. This finding not only justifies our presentation of **TIT-Score-LLM** as the SOTA instantiation of our framework but also highlights the value of our standard **TIT-Score**: it achieves performance comparable to a top-tier closed-source LLM using only a lightweight, open-

Table 4: Ablation study on LPG-Bench. The table compares the Pairwise Accuracy (%) of our standard TIT-Score against several variants, including LMM scoring and using different LLMs as evaluators. Q: Qwen3, G: Gemini-2.5-pro. For each row, red denotes the best value and blue denotes the second-best.

| VLM Backbone | Pairwise Accuracy (%) | | | | | |
|---|---|---|---|---|---|---|
| | LMM Score | Self-Eval | TIT-Score | TIT-Score-Q 8B | TIT-Score-Q 14B | TIT-Score-G |
| InternVL 3.5 8B (Wang et al., 2025c) | 36.74 | 44.07 | 60.20 | 55.61 | 59.92 | 65.58 |
| InternVL 3.5 14B (Wang et al., 2025c) | 25.10 | 49.47 | 61.27 | 55.65 | 59.83 | 66.15 |
| InternVL 3.5 30B-A3B (Wang et al., 2025c) | 42.89 | 36.59 | 59.31 | 54.30 | 59.97 | 63.99 |
| InternVL 3.5 GPT-OSS (Wang et al., 2025c) | 25.67 | 37.36 | 62.09 | 54.89 | 58.88 | 65.23 |
| Gemma 3n E4B (Gemma Team et al., 2025) | 22.07 | 49.10 | 64.14 | 53.38 | 60.94 | 64.97 |
| Qwen2.5vl 7B (Bai et al., 2025) | 9.38 | 36.74 | 63.58 | 54.68 | 61.20 | 66.41 |
| Qwen2.5vl 32B (Bai et al., 2025) | 4.10 | 40.63 | 63.14 | 57.13 | 61.98 | 66.51 |
| Mimovl 7B 2508 (Yue et al., 2025) | 33.98 | 45.71 | 64.73 | 55.36 | 61.18 | 64.42 |
| Gemini 2.5 pro (google, 2025) | 56.51 | 65.58 | 66.38 | 56.48 | 62.69 | 65.58 |

| T2I Model | Seedream3.0 | Bagel | Omnigen2 | T2I Model | Qwen-Image | gpt-image-1 | FLUX1-Dev |
|---|---|---|---|---|---|---|---|
| **Prompt:** Imagine a mystical figure known as the Wind Whisperer, standing atop a cliff overlooking an endless, rolling landscape where the sky meets the earth. This person is cloaked in flowing garments … | | | | **Prompt:** In a neon-lit, bustling cyberpunk metropolis, a young woman known as Little Red Riding Hood strides through the crowded streets. She is not the traditional fairy tale character but a modern courier, dressed in a sleek, red, hooded jacket … | | | |
| **Human(Rank)** | 1 | 8 | 13 | **Human(Rank)** | 1 | 6 | 9 |
| **TIT-Score** | 0.7717 | 0.7192 | 0.7041 | **TIT-Score** | 0.7532 | 0.6919 | 0.6853 |
| VQA-Score | 0.9092 | 0.9230 | 0.8933 | BLIPv2Score | 0.9997 | 0.9995 | 0.9995 |
| ClipScore | 0.2461 | 0.2906 | 0.2610 | Imagereward | 1.6056 | 1.0398 | 1.4363 |

Figure 4: The qualitative case study for two prompts. For both the first and second groups, TIT-Score was the only metric to correctly predict the ranking of all three images. Moreover, its numerical scores provide a clearer indication of the qualitative gap between each image compared to other models.

source embedding model. This makes **TIT-Score** a more generalizable, economical, and reliable evaluation framework, offering a more practical solution for the research community. For a more detailed analysis of evaluator and embedding model choices, please see Appendix A.9 and A.10.

## 5.4 QUALITATIVE ANALYSIS

To intuitively demonstrate the effectiveness of our TIT-Score in evaluating long-prompt adherence, we analyzed two typical examples. In the first example in 4, our score is highly consistent with the human subjective ranking, successfully placing the best-performing model at the top with a ranking that aligns with human judgment. In contrast, other existing metrics (ClipScore (Hessel et al., 2021) and VQA-Score (Li et al., 2024)) failed to accurately reflect human preferences, highlighting their limitations when handling complex prompts. In another case shown in 4, our method successfully identified subtle differences in how models adhere to prompt details, assigning a higher score to the model that was more faithful to the prompt, a result unanimously confirmed by human subjective ranking. These analyses strongly prove that our method is capable of capturing not only macroscopic quality but also performing precise evaluations at the detail level.

## 6 CONCLUSION

This paper addresses the challenge of evaluating the adherence of advanced Text-to-Image (T2I) models to long, complex prompts. To this end, we introduce LPG-Bench, a new benchmark comprising 200 meticulously designed long prompts with comprehensive human-ranked annotations. Concurrently, we propose a novel zero-shot evaluation framework that decouples visual perception from textual semantic alignment, presenting two key instantiations: the efficient, embedding-based **TIT-Score** and the peak-performance, LLM-based **TIT-Score-LLM**. Experiments on LPG-Bench reveal that most existing metrics correlate poorly with human preferences, while our framework shows superior performance and high consistency with human judgment. Specifically, **TIT-Score-LLM** achieves a 7.31% absolute improvement in pairwise accuracy over the strongest baseline, while the standard **TIT-Score** provides a highly accessible alternative with near-SOTA performance.

ETHICS STATEMENT

Our work adheres to the ICLR Code of Ethics. This work introduces LPG-Bench, a new benchmark for evaluating the long-prompt alignment of Text-to-Image (T2I) models, and TIT-Score, a novel evaluation framework to foster the development of T2I models with deeper textual understanding. The long prompts in LPG-Bench were meticulously crafted through a multi-stage pipeline involving LLM generation and manual refinement to ensure quality and diversity. All images in LPG-Bench were generated by publicly available state-of-the-art T2I models. The human annotation was conducted by trained annotators using a rigorous pairwise comparison and review mechanism to ensure reliability. Our benchmark is intended purely for academic research and evaluation, and does not involve animal subjects, medical data, or applications in high-risk domains. We have carefully considered potential societal impacts, including the risks of misuse, and aim to provide the community with a transparent and robust tool for assessing and advancing T2I models, while promoting the responsible stewardship of AI research.

REPRODUCIBILITY STATEMENT

We have made efforts to ensure the reproducibility of our work. The main paper provides a detailed description of the construction process of LPG-Bench, including prompt curation, model selection, and image generation, as outlined in Section 3. We also clearly present the design of our proposed TIT-Score framework in Section 4 and the comprehensive experimental setup in Section 5. In the Appendix, we provide further details on the long-prompt generation framework (Appendix A.3), TIT-Score implementation (Appendix A.4), and the human annotation mechanism (Appendix A.11) to facilitate replication. All resources, including the LPG-Bench benchmark and our evaluation framework, will be made publicly available upon publication. Together, these resources aim to provide sufficient transparency and guidance to reproduce our experimental results.

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

# A APPENDIX

## A.1 USE OF LLMS

During the preparation of this manuscript, we utilized a large language model (LLM) to assist with language polishing, grammar correction, and improving the clarity of expression. The core ideas, experimental design, results, and conclusions presented in this work are entirely our own.

## A.2 SIGNIFICANCE THRESHOLD CALCULATION

To determine at what level a pairwise accuracy score on our LPG-Bench benchmark can be considered statistically significant—that is, performing better than random chance—we conducted the following statistical significance analysis.

Our evaluation consists of $n = 12,832$ independent pairwise comparisons. For each pair, the judgment of an evaluation metric can be treated as a Bernoulli trial (either correct or incorrect). Our null hypothesis, $H_0$, is that the metric is performing random guessing. Under this assumption, the probability of a single success is $p = 0.5$.

Under the null hypothesis, the total number of successes, $k$, follows a Binomial distribution $B(n, p)$. Given the large number of trials $n$, this Binomial distribution can be well-approximated by a Normal distribution $\mathcal{N}(\mu, \sigma^2)$ according to the Central Limit Theorem, where:

- The mean is $\mu = n \times p = 12,832 \times 0.5 = 6,416$.
- The standard deviation is $\sigma = \sqrt{n \times p \times (1-p)} = \sqrt{12,832 \times 0.5 \times 0.5} = \sqrt{3208} \approx 56.64$.

We aim to find a minimum number of successes, $k_{min}$, such that the probability of observing $k \geq k_{min}$ (the p-value) is less than a significance level $\alpha$. We calculated the thresholds for two commonly used significance levels:

**95% Confidence (= 0.05):** For a one-tailed test, the Z-score corresponding to 95% confidence is approximately 1.645. The minimum required number of successes is therefore:

$$k_{min} = \mu + 1.645 \times \sigma \approx 6,416 + 1.645 \times 56.64 \approx 6509.17$$

This implies that at least 6510 correct judgments are needed. The corresponding accuracy threshold is:

$$\text{Accuracy Threshold} = \frac{6510}{12,832} \approx 50.73\%$$

**99.9% Confidence (= 0.001):** For a stricter 99.9% confidence level, the one-tailed Z-score is approximately 3.09. The minimum required number of successes is:

$$k_{min} = \mu + 3.09 \times \sigma \approx 6,416 + 3.09 \times 56.64 \approx 6591.02$$

This implies that at least 6592 correct judgments are needed. The corresponding accuracy threshold is:

$$\text{Accuracy Threshold} = \frac{6592}{12,832} \approx 51.37\%$$

**Conclusion:** Based on our calculations, for the 12,832 comparison pairs, a metric's pairwise accuracy must exceed **50.73%** to be considered statistically significant with 95% confidence. To reach a high confidence level of 99.9%, the accuracy needs to be above **51.37%**.

## A.3 STRUCTURED GENERATION FRAMEWORK FOR LONG PROMPTS

To ensure that each long prompt in LPG-Bench is rich, consistent, and challenging, we designed and followed a structured generation framework. This framework aims to expand a simple core theme into a detailed textual description exceeding 250 words, incorporating multi-dimensional visual elements. The core of our process is centered around the following six key building blocks:

1. **Core Subject:** A detailed depiction of the main characters, creatures, or objects in the scene. This includes their appearance, attire, material texture, emotional state, and interaction with the environment.

2. **Environment/Setting:** A depiction of the specific physical space where the subject is located. We focus on both macroscopic features (e.g., a futuristic city, an enchanted forest) and microscopic details (e.g., moss on the ground, particles floating in the air) to build an immersive world.

3. **Composition & Framing:** A definition of the visual structure and narrative perspective of the image. This part of the instruction specifies the viewing angle (e.g., wide-angle, close-up, bird's-eye view), subject placement (e.g., centered, rule of thirds), and depth of field to guide the model toward generating a cinematic composition.

4. **Lighting & Color Palette:** The establishment of the overall lighting atmosphere and color palette. The instruction details the type of light source (e.g., soft morning glow, cool neon light), its direction, and intensity, as well as the dominant color tones (e.g., warm vintage colors, high-contrast cyberpunk palette), which is crucial for creating mood and emotion.

5. **Art Style:** A clear specification of the desired art style for the final image. We cover a wide range of styles, from classical (e.g., Baroque oil painting) to modern digital art (e.g., concept art, 3D rendering), aiming to test the model's expressiveness across different artistic domains.

6. **Details & Mood:** The addition of extra details that enhance the image's texture, depth, and emotional resonance. This might include dynamic elements (e.g., fluttering cloth, falling raindrops), textural details (e.g., scratches on metal, skin texture), and the overall mood to be conveyed (e.g., tranquility, tension, epicness).

In practice, we synthesize the details conceived around these six modules into a single, fluid, and coherent paragraph of natural language. This approach ensures that each prompt functions as a self-contained, structurally complete, and detail-rich micro-storyboard, thereby posing a valid and comprehensive challenge to the long-prompt understanding capabilities of Text-to-Image models.

### A.4   TIT-SCORE IMPLEMENTATION DETAILS

#### A.4.1   PROMPT FOR VLM DESCRIPTION GENERATION

In the first stage of the TIT-Score pipeline, where a Vision-Language Model (VLM) translates image content into a textual description, we used a concise and explicit instruction. The prompt provided to all VLMs was as follows:

```
Please provide a detailed, single-paragraph
description of the image in English, using between
250 and 350 words.
```

The word count requirement of 250-350 words was chosen for two primary reasons. First, our goal was to have the VLM-generated descriptions be comparable in length to the original prompts in LPG-Bench, which average over 250 words. This ensures a degree of informational parity between the two texts being compared. Second, we observed that large language models often produce outputs that are slightly shorter than the specified upper limit when given a word count constraint. Therefore, setting a slightly higher range (250-350 words) heLPGed ensure that the resulting descriptions consistently met our desired baseline of approximately 250 words.

#### A.4.2   EMBEDDING MODEL SELECTION

For the second stage of TIT-Score, which involves calculating the similarity between text vectors, we selected the **Qwen3-Embedding** model. This decision was based on two main factors:

1. **Performance:** According to relevant public leaderboards, Qwen3-Embedding is one of the top-performing text embedding models available in the open-source domain. It demonstrates a strong capability for capturing the deep semantic information within long texts.

2. **Deployment Efficiency:** The model has a size of 8B parameters. This offers an excellent balance between high performance and practical deployability, allowing for efficient local deployment and integration into our evaluation pipeline.

## A.5 AVERAGE LEADERBOARD ANALYSIS

To more intuitively illustrate the discrepancies in the ranking capabilities of various evaluation methods on long-prompt tasks, this appendix presents the average rankings of the 13 T2I models on the LPG-Bench dataset, as determined by major evaluation metrics. The results are summarized in Table 3, which aggregates outcomes across all 200 long prompts to compute an average rank for each model under different evaluation frameworks, benchmarked against the "gold standard" of human rankings.

The following key observations can be drawn from Table 5:

1. **High Consistency of TIT-Score with Human Preferences**: Our proposed TIT-Score (using Gemini 2.5 Pro as the VLM backbone) produces a model leaderboard that demonstrates a very high degree of consistency with the rankings from human experts. Although it swaps the positions of the second and third-ranked models compared to human preference, it accurately identifies the top-tier performers and maintains a highly similar trend for ranking mid- and lower-tier models. This visually confirms TIT-Score's strong macroscopic ability to capture human judgment on long-prompt adherence.

2. **Performance of Other Advanced Methods**: As one of the strongest baselines, LMM4LMM also provides a ranking with some reference value, particularly in identifying top-performing models. However, its ability to differentiate between mid-tier models begins to deviate. In contrast, the VQA-Score ranking shows significant divergence from human preferences, suggesting that its question-answering-based approach to detail verification may not effectively measure a model's overall adherence to the comprehensive semantics and complex instructions in long prompts.

3. **Limitations of Traditional Metrics**: Traditional metrics such as ClipScore, Blipv2Score, and FGA blip2 yield rankings that show little to no correlation with expert human judgments, appearing largely unsystematic. This further substantiates our conclusion from the main text that embedding models pre-trained on short-text image pairs face severe limitations when evaluating information-rich and highly detailed long prompts.

In summary, this average leaderboard not only offers a clear snapshot of the overall capabilities of various T2I models on long-prompt generation tasks but also macroscopically validates the superior performance and reliability of our proposed TIT-Score as a next-generation evaluation metric aligned with human judgment.

Table 5: Average Leaderboard2

| Model Name | Human | TIT-Score | LMM4LMM | VQA-Score | ClipScore | Blipv2Score | FGA blip2 |
|---|---|---|---|---|---|---|---|
| Qwen-Image | 1 | 1 | 1 | 1 | 1 | 1 | 2 |
| Gemini-2.5-flash-image | 2 | 3 | 8 | 3 | 9 | 12 | 3 |
| gpt-image-1 | 3 | 2 | 3 | 2 | 4 | 5 | 8 |
| Kolors 2.1 | 4 | 5 | 4 | 12 | 13 | 9 | 11 |
| FLUX.1-Krea-dev | 5 | 4 | 6 | 4 | 6 | 2 | 4 |
| SeeDream3.0 | 6 | 6 | 12 | 5 | 12 | 7 | 10 |
| Bagel | 7 | 7 | 2 | 7 | 8 | 3 | 7 |
| Cogview4 | 8 | 10 | 5 | 9 | 10 | 11 | 12 |
| FLUX.1-Dev | 9 | 12 | 9 | 13 | 11 | 6 | 9 |
| Omnigen2 | 10 | 8 | 7 | 11 | 7 | 8 | 13 |
| Infinity | 11 | 9 | 10 | 6 | 5 | 13 | 5 |
| SD3.5 | 12 | 11 | 11 | 8 | 2 | 4 | 1 |
| Kolors | 13 | 13 | 13 | 10 | 3 | 10 | 6 |
| SRCC | 1 | **0.929** | 0.676 | 0.626 | -0.159 | 0.302 | 0.110 |

A.6  PROMPT THEME CLASSIFICATION

To demonstrate the diversity and comprehensive coverage of our LPG-Bench dataset, we developed a multi-dimensional classification system for the 200 core themes. Given that many themes are fusions of multiple concepts (e.g., a "cyberpunk museum for the Terracotta Army" combines historical and futuristic elements), a single label is insufficient. Therefore, we adopted a system comprising a primary theme category and a set of secondary descriptive tags.

A.6.1  PRIMARY THEME CLASSIFICATION

Each prompt was assigned one or more primary classifications from the six categories below:

- **Sci-Fi & Future:** This category includes concepts related to space exploration, cyberpunk, artificial intelligence, robotics, time travel, advanced technology, and post-apocalyptic settings.
  - *Examples: A lone astronaut cultivating a garden in an abandoned space station; A vast fleet of starships warping into an uncharted galaxy.*
- **Fantasy & Mythology:** This category encompasses magic, deities, mythological creatures (e.g., dragons, elves, vampires), legendary tales, and high-fantasy worlds.
  - *Examples: The coronation of a deep-sea dragon monarch; A Valkyrie observing a battlefield from the Bifrost bridge.*
- **History & Culture:** This category is rooted in specific historical periods (e.g., Victorian Era, Renaissance), cultural contexts (e.g., Ancient Egypt, Heian-period Kyoto), or cultural symbols.
  - *Examples: A grand parade of steam-powered automatons in Victorian London; A time traveler accidentally leaving a smartphone in Renaissance Florence.*
- **Surreal & Abstract:** This category emphasizes dream-like scenarios, non-Euclidean logic, the personification of abstract concepts (e.g., "Entropy"), and artistically imaginative scenes that defy physical laws.
  - *Examples: A city constructed entirely from clocks and gears; An oasis co-designed by Salvador Dalí and M.C. Escher.*
- **Nature & Ecology:** This category focuses on natural landscapes, unique ecosystems, the relationship between humanity and nature, and anthropomorphized natural elements.
  - *Examples: A celestial whale carrying a thriving ecosystem on its back; A polar city carved from the frozen skeletons of ancient behemoths.*
- **Urban & Daily Life:** This category captures scenes from urban environments or relatable everyday situations, often with a whimsical or creative twist.
  - *Examples: A cozy, ancient bookstore filled with cats on a rainy afternoon; An automated sushi bar where both chefs and patrons are robots.*

A.6.2  SECONDARY TAGS

To provide finer-grained detail, we also applied secondary tags to describe specific elements within each theme. A single prompt can have multiple tags. Examples include:

- **Art Style:** Cyberpunk, Steampunk, Biopunk, Gothic, Baroque, Film Noir.
- **Mood/Atmosphere:** Epic, Serene, Mysterious, Solitude, Dystopian.
- **Core Elements:** Robot, Architecture, Combat, Festival, Exploration.

**Application Example**  A theme such as *"The Terracotta Army of Qin Shi Huang is reactivated in a cyberpunk museum"* would be classified as follows:

- **Primary Classification(s):** History & Culture, Sci-Fi & Future.
- **Secondary Tag(s):** Cyberpunk, Robot, Urban.

### A.7 Linguistic Characteristics of Prompts

To provide an intuitive overview of the linguistic features of our long-prompt collection, we generated a word cloud from all 200 prompts. As shown in Figure 5, the visualization highlights the prominence of descriptive keywords related to visual elements such as *scene, light, color, style,* and *texture*. This reflects our structured generation framework, which emphasizes rich, multi-faceted visual instructions.

### A.8 Additional Qualitative Examples from LPG-Bench

To further illustrate the complexity and diversity of the prompts within LPG-Bench, this section provides additional qualitative examples. Each example showcases the complete long-form prompt alongside a selection of images generated by different models, demonstrating varying degrees of adherence to the detailed instructions. These cases span multiple genres, including science fiction, fantasy, and surrealism, highlighting the benchmark's capacity to test T2I models across a wide spectrum of creative challenges.

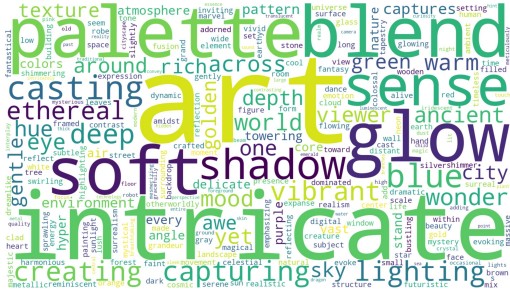

Figure 5: Wordcloud of LPG-Bench.

**Prompt**: In the vast emptiness of space, an isolated astronaut stands as the sole guardian of an abandoned space station. This astronaut is clad in a worn and slightly tattered space suit, indicative of the many missions they have undertaken and the countless hours spent in the cosmos. The helmet's visor reflects the distant glow of a nearby star, hinting at the vast universe that surrounds them. Inside the otherwise desolate space station, the astronaut tends to a burgeoning garden, a vibrant oasis of life amidst the cold steel and forgotten technology. The garden is filled with a variety of Earth's flora, from colorful daisies and tulips to exotic orchids and lush ferns, each plant a testament to resilience and hope. The scene is set within the station's large observation deck, its massive windows offering a breathtaking view of the galaxy beyond. The composition centers on the astronaut, positioned to the right of the frame, with the garden sprawling to the left, bathed in the soft, warm light of the artificial lamps hanging overhead. The scene is painted in a rich color palette of greens, purples, and yellows, contrasting with the metallic blues and grays of the station. An ethereal ambiance pervades, with the distant hum of the space station's machinery serving as a gentle backdrop. The art style is hyper-realistic, with meticulous attention to detail in the textures of both the astronaut's suit and the delicate petals of the flowers. This visual narrative conveys a sense of solitude and introspection, capturing the quiet determination of the lone astronaut nurturing life in the void of space.

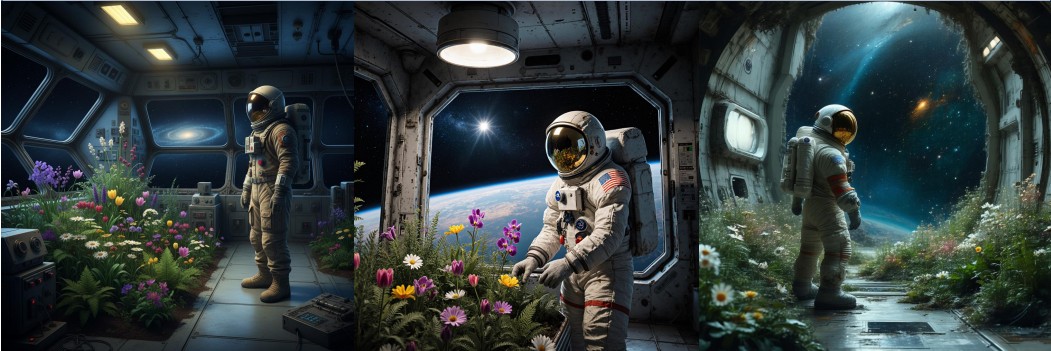

Figure 6: **Prompt:** In the vast emptiness of space, an isolated astronaut stands as the sole guardian of an abandoned space station. This astronaut is clad in a worn and slightly tattered space suit, indicative of the many missions they have undertaken and the countless hours spent in the cosmos. The helmet's visor reflects the distant glow of a nearby star, hinting at the vast universe that surrounds them. Inside the otherwise desolate space station, the astronaut tends to a burgeoning garden, a vibrant oasis of life amidst the cold steel and forgotten technology. The garden is filled with a variety of Earth's flora, from colorful daisies and tulips to exotic orchids and lush ferns, each plant a testament to resilience and hope. The scene is set within the station's large observation deck...

**Prompt**: Imagine a grand coronation ceremony of the deep sea dragon clan, set in an otherworldly underwater palace. At the heart of this underwater realm, the dragon sovereign sits majestically upon a throne crafted entirely from luminous corals that emit a soft, ethereal glow. These corals, in shades of brilliant blues, greens, and purples, illuminate the surrounding waters, casting a magical aura over the scene. The dragon sovereign is an awe-inspiring figure, its scales shimmering like liquid metal in the dappled light, with eyes like glowing sapphires that radiate wisdom and strength. The dragon wears a regal crown adorned with pearls and sea gems, symbolizing its dominion over the ocean depths. Surrounding the throne, an assembly of dragon kin and various sea creatures gather, their forms gracefully drifting in the gentle currents. The palace walls, made of opalescent shells and golden sand, are adorned with intricate carvings depicting ancient sea myths. The composition is focused, drawing the viewer's eye towards the central figure of the dragon sovereign, yet allowing deep glimpses into the vibrant life that populates this underwater kingdom. Soft beams of sunlight pierce the water from above, adding a divine luminescence to the setting. The color palette is rich and vibrant, with deep ocean hues complemented by the shimmering corals. The artistic style is a blend of realism and fantasy, capturing the surreal beauty and majesty of this oceanic realm. The overall mood is one of reverence and grandeur, as the deep sea dragon clan celebrates their new ruler in a breathtakingly beautiful ceremony.

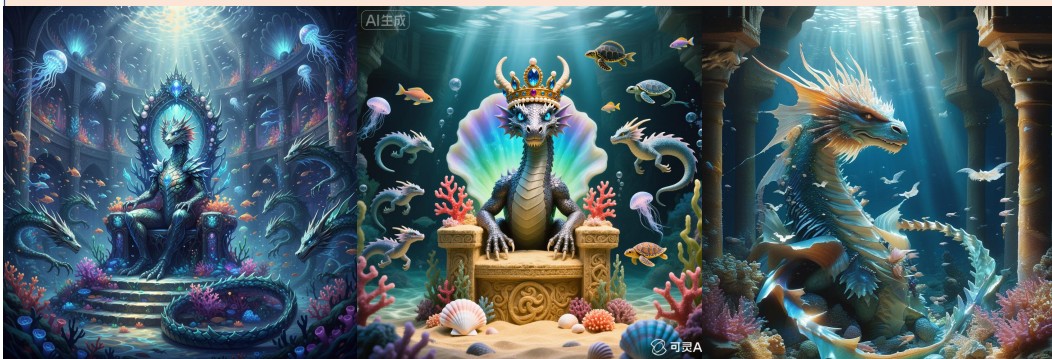

Figure 7: **Prompt:** Imagine a grand coronation ceremony of the deep sea dragon clan, set in an otherworldly underwater palace. At the heart of this underwater realm, the dragon sovereign sits majestically upon a throne crafted entirely from luminous corals that emit a soft, ethereal glow. These corals, in shades of brilliant blues, greens, and purples, illuminate the surrounding waters, casting a magical aura over the scene. The dragon sovereign is an awe-inspiring figure...

**Prompt**: Imagine a sprawling cityscape entirely constructed from the intricate mechanisms of clocks and gears. Towering structures reach towards the sky, each building formed from the elegant interlocking of brass and steel cogs, with clock hands sweeping grand arcs across their surfaces. The streets below are paved with the finely polished faces of enormous clocks, each one ticking in perfect harmony, creating a mesmerizing symphony of time. In the city center stands a colossal clock tower, its face a kaleidoscope of spinning wheels and oscillating pendulums, casting a shadow that slowly creeps across the city as time passes. The atmosphere is a blend of metallic coolness and warmth from the golden hues of the setting sun, casting long, dramatic shadows and bathing everything in a soft, orange glow. The composition captures a wide panoramic view, showcasing the city's grandeur and complexity, with a focus on the interplay of light and shadow across the mechanical landscape. The color palette is a rich tapestry of golds, bronzes, and silvers, punctuated by the deep blues of the sky and the occasional flash of crimson from a particularly striking gear. The art style is a blend of steampunk and realism, with meticulous attention to detail in the depiction of each gear and mechanism. This city is not just a marvel of engineering, but an embodiment of time itself, evoking a sense of wonder and introspection as you ponder the relentless passage of time and the intricate dance of seconds, minutes, and hours.

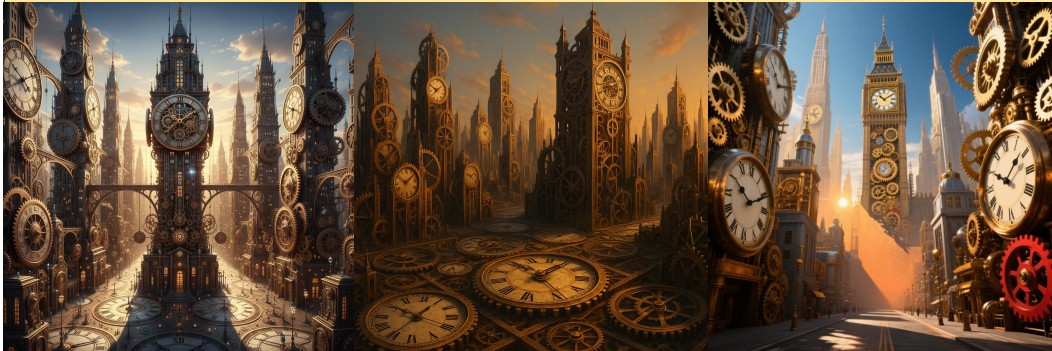

Figure 8: **Prompt:** Imagine a sprawling cityscape entirely constructed from the intricate mechanisms of clocks and gears. Towering structures reach towards the sky, each building formed from the elegant interlocking of brass and steel cogs, with clock hands sweeping grand arcs across their surfaces. The streets below are paved with the finely polished faces of enormous clocks...

**Prompt**: In a scene of breathtaking splendor, a formidable Norse Valkyrie stands poised at the edge of the legendary Bifröst, also known as the Rainbow Bridge, her silhouette framed against the vibrant spectrum of colors that arc across the sky. She is clad in resplendent armor that glistens with the hues of silver and azure, intricately adorned with runic inscriptions that seem to pulse with an inner light. Her long, golden hair cascades like a river of sunshine, catching the ethereal glow of the bridge. Her gaze is intense and noble, surveying the vast expanse of the human battlefield below, where warriors clash in a dance of chaos and valor. The environment is a juxtaposition of heavenly and earthly realms: above, the skies shimmer with the iridescent lights of Asgard, while below, the earth is shrouded in the tumultuous fog of war, the air thick with the cries of battle and the clang of steel. The composition captures a dynamic perspective, as if the viewer is witnessing this scene from slightly above and behind the Valkyrie, sharing her divine vantage point. The lighting is a dramatic interplay of celestial radiance and shadow, casting a mystical aura over the scene. The color palette is a rich tapestry of luminescent pastels and dark, brooding tones that accentuate the epic narrative. The artwork is rendered in a majestic Norse mythology style, reminiscent of classical paintings yet infused with a modern cinematic flair. Details such as the Valkyrie's expression of solemn duty and the intricate design of her armor convey a profound sense of mythic grandeur and emotional depth, inviting the viewer into a world where gods and mortals meet.

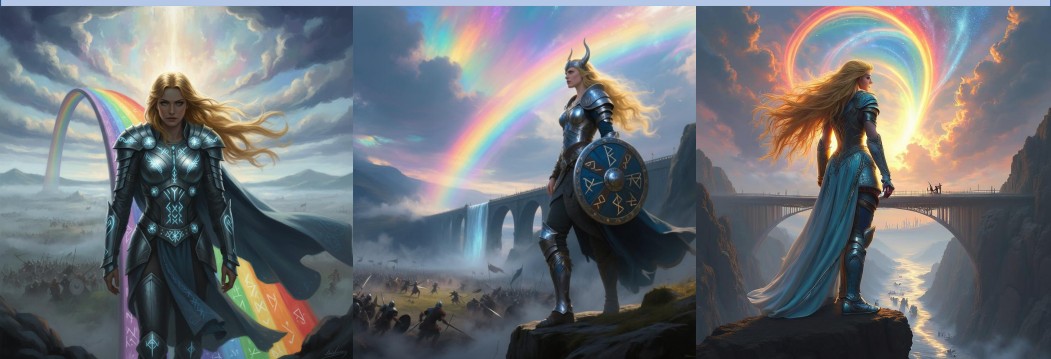

Figure 9: **Prompt:** In a scene of breathtaking splendor, a formidable Norse Valkyrie stands poised at the edge of the legendary Bifröst, also known as the Rainbow Bridge, her silhouette framed against the vibrant spectrum of colors that arc across the sky. She is clad in resplendent armor that glistens with the hues of silver and azure, intricately adorned with runic inscriptions that seem to pulse with an inner light...

**Prompt**: In a world where all colors have vanished, imagine a scene of profound desolation and quiet beauty. A young child, clad in simple grey garments, stands at the heart of this colorless world. The child is small and slender, perhaps seven or eight years old, with a look of both curiosity and wonder etched upon their features. They have stumbled upon a miraculous anomaly: a single, vivid red flower blooming amidst the monochrome landscape. The flower's petals are a striking contrast to the world around it, its intense crimson hue pulsating with life as if defying the bleakness of its surroundings. The setting is a vast open field, stretching endlessly under a pale, washed-out sky. The ground is covered with a fine layer of ash-like dust, swirling occasionally with the whisper of the wind. In the distance, faint outlines of leafless trees can be seen, their branches like skeletal fingers reaching skyward. The composition centers the child and the flower, capturing a moment of stillness and awe. The lighting is soft and diffused, casting gentle shadows that emphasize the flower's vivid color. The overall palette is muted, dominated by shades of grey and white, with the occasional hint of shadow. This makes the red flower all the more striking, as it seems to glow with an inner light, drawing the viewer's eye irresistibly towards it. The art style should reflect a blend of hyperrealism and surrealism, capturing the stark reality of a world devoid of color, yet infused with the surreal magic of the lone red flower. The mood is one of poignant beauty and hope, as the flower symbolizes resilience and life in a world otherwise drained of vibrancy. Details such as the texture of the flower's petals, the expression on the child's face, and the subtle interplay of light and shadow are crucial to conveying the depth and emotion of the scene.

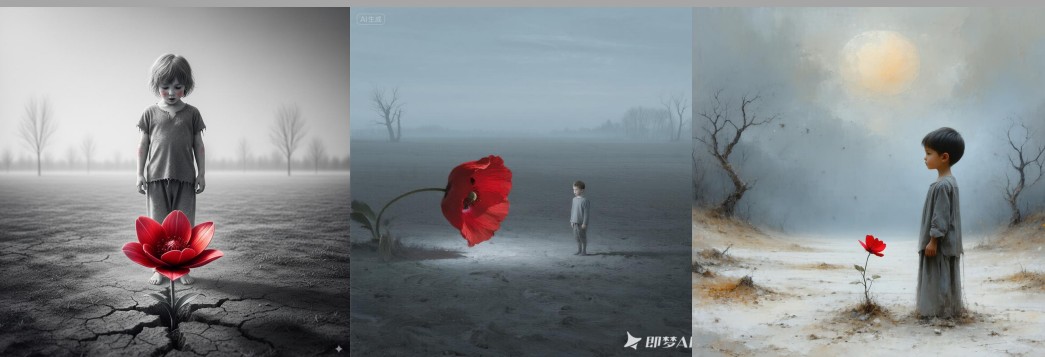

Figure 10: **Prompt:** In a world where all colors have vanished, imagine a scene of profound desolation and quiet beauty. A young child, clad in simple grey garments, stands at the heart of this colorless world. The child is small and slender, perhaps seven or eight years old, with a look of both curiosity and wonder etched upon their features. They have stumbled upon a miraculous anomaly...

## A.9 Considerations on the
## Evaluator Choice: Why the Standard TIT-Score Utilizes an Embedding Model

In our ablation study, a noteworthy result is that when a top-tier Large Language Model like Gemini 2.5 Pro is used to directly score the semantic similarity between the VLM-generated description and the original prompt (i.e., Variant 3), its evaluation accuracy is nearly comparable to, and at times even slightly surpasses, our standard TIT-Score. This finding is interesting in itself, and structurally, this "VLM description + LLM evaluation" paradigm is also an implementation of the TIT-Score's decoupled philosophy, which could be considered TIT-Score2. However, the standard method we ultimately advocate and propose in this paper is the use of a standalone, open-source embedding model for the final similarity calculation. This decision is based on a comprehensive consideration of the method's generalizability, accessibility, and robustness.

One of our core objectives in proposing TIT-Score is to create an evaluation framework that is widely applicable to the research community, convenient to deploy, and reliable in its results. Although using a top-tier LMM for evaluation performs exceptionally well in specific cases, this success exhibits a certain "fragility" and model-dependency. As our experiments revealed, when we applied a precise-scoring instruction to other LMMs, they often exhibited a "pseudo-precision" phenomenon, proving that this end-to-end scoring capability is not a widespread feature among models but rather a unique ability of a few top-tier ones. Building the core component of an evaluation framework upon such a rare and typically closed-source capability would severely limit its generalizability. In contrast, dedicated text embedding models (like our chosen Qwen3-Embedding) are optimized to reliably measure text similarity within a stable, high-dimensional semantic space. They have lower computational requirements, can be easily deployed locally, and their performance is predictable and stable, independent of complex prompt engineering. Therefore, for the sake of the method's pervasiveness and robustness, we choose the embedding-based approach as the standard implementation of TIT-Score, as it provides a more reliable, economical, and generalizable solution for evaluating long-prompt-to-image alignment.

## A.10 Sensitivity Analysis on Embedding Model Choice

In our TIT-Score framework, the choice of the text embedding model is a critical component for achieving high-fidelity semantic alignment. For the experiments presented in the main body of this paper, we selected the `Qwen3-Embedding-8B` model. This decision was primarily based on its state-of-the-art performance on public text embedding benchmarks (e.g., MTEB), where it demonstrated a strong capability for capturing the deep semantics of long-form text.

However, to validate the robustness of the TIT-Score framework and investigate its sensitivity to the parameter scale of the embedding model, we conducted an additional comparative experiment in this appendix. We maintained the identical TIT-Score pipeline but replaced the embedding model with a `4B` parameter version. We then re-evaluated the performance across all VLM backbones on LPG-Bench.

The following tables present the complete results for TIT-Score across all evaluation metrics, using the 8B embedding model (Table 6) and the 4B embedding model (Table 7), respectively.

Table 6: Performance of TIT-Score with the 8B Embedding Model (`Qwen3-Embedding-8B`)

| VLM Backbone (TIT-Score) | Acc (%) | SRCC | KRCC | nDCG |
|---|---|---|---|---|
| InternVL 3.5 8B | 60.20 | 0.2496 | 0.1984 | 0.8033 |
| InternVL 3.5 14B | 61.27 | 0.2826 | 0.2183 | 0.8034 |
| InternVL 3.5 30B | 59.31 | 0.2323 | 0.1813 | 0.7960 |
| InternVL 3.5 GPT-OSS | 62.09 | 0.2988 | 0.2329 | 0.8026 |
| Gemma 3n E4B | 64.14 | 0.3468 | 0.2722 | 0.8101 |
| Qwen2.5-VL 7B | 63.58 | 0.3280 | 0.2584 | 0.8111 |
| Qwen2.5-VL 32B | 63.14 | 0.3208 | 0.2509 | 0.8106 |
| Qwen2.5-VL 72B | 63.12 | 0.3189 | 0.2516 | 0.8085 |
| MiMo-VL 7B | 64.73 | 0.3595 | 0.2810 | 0.8135 |
| GPT-4o | 64.64 | 0.3630 | 0.2804 | 0.8209 |
| Gemini 2.5 Pro | **66.38** | **0.3953** | **0.3099** | **0.8344** |

Table 7: Performance of TIT-Score with the 4B Embedding Model

| VLM Backbone (TIT-Score) | Acc (%) | SRCC | KRCC | nDCG |
|---|---|---|---|---|
| InternVL 3.5 8B | 60.16 | 0.2555 | 0.1981 | 0.8043 |
| InternVL 3.5 14B | 61.81 | 0.2944 | 0.2287 | 0.8022 |
| InternVL 3.5 30B | 59.45 | 0.2365 | 0.1833 | 0.7960 |
| InternVL 3.5 GPT-OSS | 62.10 | 0.2997 | 0.2338 | 0.7990 |
| Gemma 3n E4B | 64.10 | 0.3483 | 0.2707 | 0.8168 |
| Qwen2.5-VL 7B | 62.91 | 0.3139 | 0.2471 | 0.8063 |
| Qwen2.5-VL 32B | 63.03 | 0.3188 | 0.2475 | 0.8203 |
| Qwen2.5-VL 72B | 63.08 | 0.3147 | 0.2514 | 0.8058 |
| MiMo-VL 7B | 64.49 | 0.3560 | 0.2781 | 0.8130 |
| GPT-4o | 64.57 | 0.3599 | 0.2800 | 0.8205 |
| Gemini 2.5 Pro | **65.62** | **0.3783** | **0.2968** | **0.8303** |

**Analysis and Conclusion**   By comparing the results from the two experiments, we can draw two primary conclusions. First, the 8B version of the embedding model demonstrates a slight performance advantage across all metrics, particularly when paired with top-tier VLMs like Gemini 2.5 Pro, which validates our initial choice. Second, and more importantly, the 4B version still achieves highly competitive results, with only a marginal decrease in performance across the board. Furthermore, the relative performance ranking of the different VLM backbones remains largely consistent between the two tables.

This finding strongly supports the robustness of the TIT-Score framework. Its superior performance is not merely contingent on a specific, large-scale embedding model but rather stems from its core "decoupled" design philosophy. This implies that researchers and developers can flexibly choose embedding models of different scales based on their available computational resources and still obtain reliable evaluation results that correlate highly with human judgment. This significantly enhances the practicality and accessibility of TIT-Score for broader applications.

### A.11   Detailed Mechanism of Human Preference Annotation

To ensure the high reliability and rigor of the human preference annotations in the `LPG-Bench` dataset, we designed and implemented a multi-stage annotation and review mechanism. We recruited a team of 15 annotators who underwent standardized training to ensure a deep understanding of the evaluation criteria for text-to-image generation models.

**1. Task Setup and Scoring Criteria**   The annotation task employed a **pairwise comparison** scheme. For any two images generated from the same long-form prompt, annotators were asked to compare them and provide one of three judgments: "Left image is better", "Right image is better", or "Tie". This relative ranking approach effectively mitigates the issues of high subjectivity and inconsistency associated with absolute rating scales.

Our core annotation criteria are structured around two levels of adherence:

1. **Macro-element Adherence:** This is the primary evaluation dimension. Annotators first check if the image successfully renders the prompt's described **core subjects** (e.g., characters, objects), **environmental settings**, and **key actions or relationships**. An image failing to include these fundamental elements is considered low-quality and is quickly disqualified.

2. **Micro-detail Adherence:** Once macro-elements are correctly rendered, annotators delve into evaluating the image's ability to reproduce the prompt's **complex details**. This includes, but is not limited to:

   - **Positional Relationships:** Assessing whether the relative positions of different objects in the image match the prompt's requirements.
   - **Logical Relationships:** Determining if the image accurately portrays the implied logic or narrative of the prompt.
   - **Artistic Style and Mood:** Examining whether the overall style and emotional atmosphere of the image are consistent with the prompt's description.

**2. Aggregation of Judgments and Final Decision** To aggregate the judgments from the 15 annotators into a single, reliable outcome for each image pair (e.g., Image A vs. Image B), we implemented a hierarchical decision-making process. For each pair, the votes were first tallied into three categories: $V_A$ (votes for Image A), $V_B$ (votes for Image B), and $V_T$ (votes for a Tie), where $V_A + V_B + V_T = 15$. The final outcome was determined by applying the following rules sequentially:

1. **Rule 1: Strong Consensus.** This rule identifies outcomes with a very high degree of agreement. A final decision is made if any party receives a supermajority of $\geq 2/3$ of the votes. Based on our 15 annotators, this threshold is 10 votes.

   - If $V_A \geq 10$, the final judgment is that A wins.
   - If $V_B \geq 10$, the final judgment is that B wins.
   - If $V_T \geq 10$, the final judgment is a Tie.

   *Rationale: This high threshold ensures that the decision is robust against a small number of outlier opinions.*

2. **Rule 2: Significant Advantage.** If no strong consensus is reached, this rule assesses whether there is a clear preference among annotators who did not vote for a tie. A decision is made if: (1) The total number of preference votes ($V_{pref} = V_A + V_B$) is at least 8, and (2) one image captures at least 75% of these preference votes.

   - If $V_{pref} \geq 8$ and $V_A/V_{pref} \geq 0.75$, the final judgment is that A wins.
   - If $V_{pref} \geq 8$ and $V_B/V_{pref} \geq 0.75$, the final judgment is that B wins.

   *Rationale: This rule honors a clear preference margin among decisive annotators, even with a notable number of "Tie" votes.*

3. **Rule 3: High Disagreement and Final Arbitration.** Any pair not resolved by the above rules is flagged as having "High Disagreement" (e.g., vote splits like 7-6-2 or 5-5-5). These cases are escalated to a final review by a panel of 3 senior experts for arbitration.

   - The final judgment is determined by the majority vote of the expert panel.
   - If the expert panel cannot reach a majority (i.e., each expert gives a different verdict), the final outcome is designated as a "Tie".

   *Rationale: This ensures that ambiguous or contentious comparisons do not introduce noise into the final ranking and are resolved by the most experienced reviewers, defaulting to a tie in cases of ultimate ambiguity.*

**3. Ranking Generation** Once a definitive result (win, loss, or tie) was established for every possible pair of images for a given prompt using the aggregation mechanism described above, we applied a mechanism inspired by **sorting algorithms** to generate a final relative ranking of all 13 models. This process constructs a complete leaderboard that accommodates ties, reflecting not only the models' average performance but also, through the count of top-ranked images, their ability to produce high-quality "peak" outputs.

## A.12 VS SHOW

This appendix provides a series of selected qualitative case studies to visually demonstrate the human annotation scale used for LPG-Bench. Each case includes a complete long-form prompt and two sets of comparative images. Our domain experts performed a pairwise comparison for each image set, using a simple notation to indicate their judgment: a "✓" signifies that the image is superior to its counterpart in text alignment, an "✗" indicates that it is inferior, and a "=" denotes that both images are of comparable quality and were judged as a "tie." These cases offer a transparent and reproducible basis for understanding the rigor and reliability of our annotated data, providing deep insights into the models' capabilities in executing complex instructions.

**Prompt**: In the dimly lit attic of an ancient house, a young child stands captivated by a peculiar map sprawled across a dusty old trunk. The child, a curious and adventurous soul with tousled hair and bright eyes, is dwarfed by the labyrinthine piles of forgotten relics and cobwebbed antiques surrounding them. The attic, with its wooden beams and small dust-laden window, is bathed in a muted, golden sunlight filtering through the cracks, casting long shadows and a warm, nostalgic glow. The focal point is the map—an intricately detailed parchment teeming with vibrant colors, strange symbols, and whimsical illustrations of mythical creatures and enchanted landscapes. The map appears to shimmer with a life of its own, inviting the child to trace the pathways with their fingers, each touch sparking a faint luminescence. The camera angle is slightly above, providing a bird's-eye view of the scene, accentuating the child's sense of wonder and discovery as they lean over the map. The color palette is a harmonious blend of earthy browns, deep greens, and touches of iridescent blues and golds, evoking a magical yet timeless atmosphere. The style is reminiscent of classic fantasy illustrations, with a touch of impressionism to add a dreamlike quality. The scene is imbued with a sense of mystery and anticipation, capturing the child's awe and the tantalizing promise of adventure in a world beyond imagination.

The successful image shows the map 'on the box' and the child has bright eyes; the failed image shows neither.

The successful image shows the map 'on the box' and the child has bright eyes; the failed image shows neither.

Figure 11: The evaluation of these two image sets is straightforward. The main subjects are a child and a map placed on a box. The failed samples in both comparison groups have two primary flaws: first, they both lack the spatial relationship of being "on the box." Second, the child's eyes are not visible, failing to represent the "bright eyes" attribute. Based on these two points, making a judgment is quite easy.

**Prompt**: Imagine a grand and awe-inspiring scene set on the fabled Olympus Mountain, where an assembly of ancient Greek philosophers engage in an animated debate with the mighty gods of the Greek pantheon. At the center of this dynamic tableau, the wise and bearded Socrates, draped in a flowing white toga, gestures passionately as he presents his arguments. To his right, Plato, with his thoughtful gaze and scroll in hand, listens intently. On the opposite side, the formidable Zeus, seated on his golden throne, his eyes flashing like lightning, responds with booming authority. Hera, beside Zeus, exudes regal grace and wisdom. Surrounding them are other philosophers like Aristotle and Pythagoras, each contributing their unique perspectives, and gods like Athena and Apollo, who watch with keen interest. The setting is a majestic stone amphitheater perched on the mountain's peak, with panoramic views of the sky ablaze in hues of orange and pink as the sun sets on the horizon. The composition captures the grandeur of the gathering, with the philosophers and gods arranged in an intricate spiral that draws the eye inward. The lighting is dramatic, casting deep shadows and highlighting the divine glow surrounding the gods, while the philosophers are bathed in the warm glow of the setting sun. The color palette is rich and vibrant, dominated by blues, golds, and earthy tones. The art style is a blend of classical realism and romanticism, evoking the timelessness and drama of the scene.

The failed image's primary issue is its lack of emphasis on the main subject: "Socrates in a white robe delivering a passionate discourse."

The failed image's completely missing essential elements like the white robe, a prominent figure, and the passionate discourse itself.

Figure 12: For the first pair, the image on the right lacks emphasis on the primary subject, which is "Socrates in a white robe delivering a passionate discourse." The assessment for the second pair is more definitive, as the failed example is devoid of essential elements: the white robe, a prominent figure, and the passionate statement itself. While neither of the winning images constitutes a perfect execution of the instruction, their performance is markedly superior to that of the losing counterparts.

**Prompt**: In the heart of the jazz age, beneath the bustling streets of 1920s New York, lies a hidden speakeasy, a clandestine haven of music, magic, and mystery. The room is dimly lit, with smoky tendrils curling under the amber glow of vintage light fixtures. In the center of this intimate, opulent space, a charismatic magician captivates the audience with his extraordinary performance. Clad in a tailored midnight blue tuxedo, adorned with a silk top hat and white gloves, he moves with the grace of a dancer, his every gesture commanding attention. The magician stands before a small stage, where a rich burgundy velvet curtain serves as a backdrop. Around him, patrons are seated at small round tables, sipping on colorful cocktails, their faces illuminated by flickering candlelight. The air is thick with the scent of aged whiskey and sweet cigars, an aroma synonymous with the era's hedonistic indulgence. As the magician waves his wand, reality bends; cards float and dance in mid-air, illusions shimmer into existence, and rabbits emerge from top hats in a burst of iridescent sparkles. The palette of the scene is a luxurious blend of deep maroons, golds, and blacks, creating an atmosphere both enchanting and slightly surreal. The style of the image evokes the works of the Art Deco movement, with its bold geometric forms and intricate details. The mood is one of wonder and awe, echoing the unbridled creativity and exuberance of the Roaring Twenties.

The inferior image is clearly identifiable as it lacks key elements like the rabbit and playing cards.

The depiction of the "rabbit" element is the clear basis for judgment.

Figure 13: The assessment for the first pair is simple, as the inferior image lacks key elements like the rabbit and playing cards. For the second pair, the distinction is less pronounced. Nevertheless, the depiction of the "rabbit" element serves as a clear basis for judging which of the two more faithfully adheres to the instruction.

**Prompt**: In the ethereal realm where the earth meets the sky, a scene of tranquil majesty unfolds. A solitary monk, clad in traditional saffron robes, stands as a timeless sentinel amidst a sea of clouds. He is poised beneath the ancient eaves of a temple that seems to float effortlessly above the world, its silhouette a serene contrast against the endless azure expanse. The monk's hands are gracefully raised, preparing to strike an ancient bronze bell that hangs suspended within an ornately carved wooden frame. The bell, imbued with centuries of history, bears intricate patterns that tell stories of devotion and peace. Surrounding the monk is a delicate veil of mist, shifting gently with the breeze, adding an air of mysticism to the scene. The composition is framed from a low angle, capturing the grandeur of the temple against the expansive sky, with the monk positioned slightly to the left, drawing the eye towards the center of the frame where the bell awaits its resonant toll. Sunlight streams through the clouds, casting soft, golden hues across the scene, highlighting the textures of the monk's robe and the fine details of the temple's architecture. The color palette is a harmonious blend of soft blues and warm golds, evoking a sense of peace and spiritual reflection. The art style is a fusion of hyper-realistic detail and subtle impressionistic touches, capturing both the physical beauty and the profound serenity of the moment…

The inferior image fails to include the "clock in a wooden frame" and the "gesture of preparing to strike."

The winning image's portrayal of the "preparing to strike" action is markedly superior to its counterpart.

Figure 14: The distinction in the first pair is pronounced, as the inferior image fails to include the "clock in a wooden frame" and the "gesture of preparing to strike." In the second pair, while the winning image also inadequately represents the "clock in a wooden frame" element, its portrayal of the "preparing to strike" action is markedly superior to that of its counterpart.

**Prompt**: In the heart of Renaissance Florence, amidst the bustling streets and towering cathedrals, a curious scene unfolds. A time traveler, dressed in a peculiar blend of futuristic and period attire, accidentally leaves behind a sleek, modern smartphone on a rustic wooden table outside a quaint café. This juxtaposition of eras is set against the backdrop of Florence's iconic architecture: the imposing silhouette of the Duomo looms in the distance, its intricate façade bathed in the warm glow of the setting sun. The cobblestone streets are alive with citizens in period attire, their expressions a mixture of bewilderment and fascination as they gather around this mysterious device, its screen flickering softly with images and lights unknown to them. The composition captures the essence of discovery and wonder, as if the viewer is peering through a window into a moment where time itself has paused. The scene is suffused with a golden, almost magical light, casting long shadows and illuminating the rich textures of the stone buildings, the vibrant hues of the people's clothing, and the glint of curiosity in their eyes. The color palette blends the earth tones of the Renaissance with the subtle modernity of the smartphone, creating a harmonious balance between past and future. The art style mirrors the detailed realism of Renaissance paintings, with a hint of contemporary digital flair to emphasize the anachronistic presence of the smartphone. Delicate details, such as the intricate carvings on the buildings and the gentle folds of fabric, enrich the scene…

The winning image effectively portrays the curious expressions of surrounding people towards the mobile phone; the losing image completely omits this key detail.

The judgment in this set hinges on depicting the onlookers' curious expressions, a detail the superior image successfully includes.

Figure 15: In both comparison pairs, the advantage of the winning image is distinct. The crucial differentiating factor is the portrayal of the surrounding people's curious expressions towards the mobile phone. Both winning images represent this element effectively; in contrast, the losing images in both groups completely fail to include this key detail.

**Prompt**: Imagine an immense, awe-inspiring interstellar library that stretches infinitely across the cosmos. This celestial edifice is not made of stone or metal, but of shimmering, ethereal structures that seem to be woven from the very fabric of the universe. The core subject of this visual spectacle is the books themselves, each with pages that are not ordinary paper, but flowing nebulae of vibrant colors. These cosmic pages shift and dance with gentle currents, the rich tapestries of stars and gossamer clouds of interstellar dust creating dynamic and ever-changing patterns. It is as if each book tells its story through the movement and glow of these celestial phenomena. The environment in which these starry tomes are housed is a vast, cavernous space, reminiscent of a grandiose hall with arching ceilings formed by swirling galaxies. The walls of this library are translucent veils of light, through which the twinkling of distant stars can be seen, giving the sense that the entire structure is floating in deep space. The composition of the scene is designed to evoke a sense of wonder and discovery, with the viewer's perspective starting at the entrance of the library, looking down a seemingly endless corridor lined with these luminous volumes. The lighting in this scene plays a crucial role, as it is entirely natural, emanating from the celestial phenomena themselves. The color palette is a rich spectrum of blues, purples, and pinks, accented by the occasional burst of gold or silver starlight, creating an otherworldly glow that fills the scene with a serene yet…

The key differentiator is whether the book pages and walls are composed of nebulas with a dynamic quality, a concept the winning image captures.

The assessment is straightforward, as the inferior image entirely lacks the primary subject of a book.

Figure 16: The prompt is rather abstract. In the first comparison, the primary point of differentiation is whether the book pages and walls are composed of nebulas and exhibit a dynamic quality. The winning image successfully captures these aspects, while the losing image fails to adequately represent this abstract concept. For the second comparison, the assessment is more straightforward: the inferior image entirely lacks the primary subject, a book.

**Prompt**: In a serene Japanese garden, where cherry blossoms are in full bloom, their delicate petals drifting gently through the air, a dramatic scene unfolds. At the heart of this picturesque setting, a seasoned samurai, clad in traditional armor, stands poised. His armor is ornate, with intricate details of dragons and ancient symbols etched into the metal. The samurai's katana, polished to a mirror-like finish, glints in the dappled sunlight filtering through the cherry blossom branches. Opposite him, a futuristic humanoid robot stands ready, its sleek metallic body reflecting the pink hues of the cherry blossoms. The robot is an amalgamation of advanced technology, with a face devoid of emotion yet carrying an aura of formidable intelligence. The garden is alive with vibrant colors, the soft pink of the cherry blossoms contrasts with the rich green of the perfectly manicured grass and ancient bonsai trees surrounding the combatants. In the background, a traditional Japanese tea house with sliding doors and wooden structures adds to the authenticity of the setting. The composition is akin to a dynamic, wide-angle shot, capturing the tension between the ancient and the futuristic, the organic and the mechanical. The lighting is natural and soft, casting gentle shadows that dance across the scene, enhancing the textures of the samurai's armor and the robot's sleek frame. The color palette is a harmonious blend of pastels and metallics, with the pinks, greens, and silvers creating a visually striking contrast. The art style is a fusion of hyper-realistic and traditional …

The losing image fails because it is missing the crucial elements of floating cherry blossom petals and the structure of a wooden sliding door.

The distinction in this set is determined by the presence of both floating cherry blossom petals and a wooden sliding door.

Figure 17: For both pairs, the primary distinction is that the losing images are missing crucial elements: floating cherry blossom petals and the structure of a wooden sliding door.

**Prompt**: In the whimsical realm of Sugaria, a world where every element is made of desserts and candies, imagine a towering castle constructed entirely out of gingerbread walls and candy cane turrets. The castle stands amidst a sprawling landscape of rolling hills made of marshmallow fluff, with rivers of molten chocolate winding their way through the valleys. The sky above is a swirl of cotton candy clouds, casting a soft pink and blue hue over the scene. As you gaze upon this sweet kingdom, notice the intricate details of the environment: gumdrop trees line the pathways, their leaves a vibrant mix of fruit-flavored jellies. Lollipop lampposts light the cobblestone streets, their sugary glow creating a magical ambience. In the foreground, a group of animated marzipan figures gather around a fountain of syrup, their expressions full of joy and wonder. The composition is a wide-angle shot, capturing the vastness of Sugaria with the castle as the central focus, while the candy landscape unfolds in the background. The lighting is reminiscent of a golden hour, warm and inviting, enhancing the pastel color palette that dominates the scene. The art style is reminiscent of a dreamy, storybook illustration, with a touch of surrealism to accentuate the fantastical nature of the world. Capturing the essence of childlike wonder and the joy of discovery, this scene invites you to indulge in a visual feast of imagination and creativity.

The images in this set are largely on par in terms of the elements they include; therefore, the comparison is ruled a draw to ensure accuracy.

A clear winner cannot be determined as both images are highly consistent in their core elements, resulting in a draw for this group.

Figure 18: Although a significant disparity exists between the two main comparison groups, the images within this particular group are largely on par in terms of the elements they include. Therefore, to maintain the highest possible accuracy in the evaluation, this specific comparison is ruled a draw.

