# OpenReview forum: "TIT-Score: Evaluating Long-Prompt Based Text-to-Image Alignment via Text-to-Image-to-Text Consistency"
_ICLR.cc/2026/Conference — ICLR 2026 Conference Withdrawn Submission_

### Official Review · Reviewer_zgR9 · 2025-10-26

**Soundness:** 3
**Presentation:** 3
**Contribution:** 2
**Rating:** 4
**Confidence:** 4

**Summary:**

The paper introduces LPG-Bench, a benchmark with 200 long prompts for evaluating text-to-image models, and proposes TIT-Score, a “text-to-image-to-text” framework that converts a generated image back to text and then measures semantic similarity with the original prompt.

**Strengths:**

1. LPG-Bench is relatively large and well-documented, with manual refinement and human preference annotations.

2. the proposed metric correlates well with human judgment and outperforms common baselines.

**Weaknesses:**

1. Novelty and conceptual depth are limited: The TIT-Score framework is essentially a straightforward pipeline combining an existing captioning VLM + text embedding similarity, a known idea explored in prior “image caption consistency” or “caption-re-caption” works (e.g., CLIP-I2T, BLIP-I2T).

2. No discussion on domain coverage (e.g., scene diversity, cultural context, abstract concepts).

3. The paper overlooks some directly relevant prior works on long or paragraph-level text-to-image generation, such as ParaDiffusion, which not only tackles long, information-rich text inputs but also includes its own long-prompt evaluation. Ignoring these studies weakens the novelty and positioning claims, as TIT-Score should be compared or at least discussed alongside such baselines.

4. Lack of generalization analysis or simple disscusion(e.g., shorter prompts, cross-domain tests, multilingual prompts).
[1] Paragraph-to-Image Generation with Information-Enriched Diffusion Model

**Questions:**

None

---

### Official Review · Reviewer_erxp · 2025-10-27

**Soundness:** 2
**Presentation:** 2
**Contribution:** 2
**Rating:** 2
**Confidence:** 4

**Summary:**

The paper introduces LPG-Bench, a benchmark specifically designed to evaluate text-to-image (T2I) generation for long prompts (averaging 250+ words). To address the limitations of existing metrics like CLIP-Score on these long prompts, the authors propose TIT-Score (Text-to-Image-to-Text). This "decoupled" framework first uses a VLM to caption the generated image, then measures the semantic similarity between this new caption and the original long prompt using either a text embedding model (TIT-Score) or an LLM (TIT-Score-LLM).

**Strengths:**

- The benchmark addresses a clear gap in current evaluation by focusing specifically on very long, narrative-style prompts (>250 words), which approach the context limits of modern commercial models.

- The "describe-then-compare" approach is a logical attempt to handle long contexts, avoiding the noise inherent in asking a VLM to directly score a complex image-text pair end-to-end.

- The authors provide a good ablation study demonstrating that end-to-end "LMM Scoring" performs poorly on these long prompts, justifying their decoupled approach.

**Weaknesses:**

- One of the main contributions of this work is proposing a new metric for evaluating text-to-image generation. However, the paper fails to cite or compare against established question-decomposition metrics that actually do what they seemed to incorrectly ascribe to VQAScore. Key missing baselines include:

  - TIFA [1]: A standard metric that generates specific questions from the prompt to verify details.

  -  DSG  [2]: Another prominent metric that uses scene graphs to generate fine-grained evaluation questions.

  - Gecko [3]: A recent benchmark that also utilizes diverse QA pairs for evaluation. Omitting these standard, directly relevant "decompose-and-verify" metrics while mischaracterizing VQAScore is a severe flaw in evaluating the state of the art.

- The authors state that "VQA-Score... verify factual details by decomposing prompts into questions" (Section 2.3). This is factually incorrect. As per Eq. 1 of the VQAScore paper, this metric typically computes the probability of a "Yes" answer given the entire prompt as a single question (e.g., "Does this image match the text: [prompt]?"), specifically to avoid the decomposition they claim it uses. This fundamental misunderstanding of a key baseline calls into question the rigor of their literature review and baseline comparisons.

- The TIT-Score metric hinges on the VLM's ability to perfectly capture a 250-word prompt in a single caption. This introduces a severe information bottleneck. If the VLM omits a detail (common in dense images) or hallucinates elements, the subsequent text similarity score is fundamentally flawed. The paper fails to account for standard VLM captioning biases or at least providing some empirical evidence that this issue won’t happen in practice.

- Lack of reporting critical details on human evaluation: While Appendix A.11 details a rigorous-sounding mechanism for aggregating human judgments, it fails to report how many of the 12,832 pairs fell into the "High Disagreement" category requiring expert arbitration. Without these statistics, it is impossible to gauge the actual reliability of the human ground truth.


[1] Hu et al. TIFA: Accurate and Interpretable Text-to-Image Faithfulness Evaluation with Question Answering, ICCV, 2023

[2] Cho et al., Davidsonian Scene Graph: Improving Reliability in Fine-grained Evaluation for Text-to-Image Generation, ICLR 2024

[3] Wiles et al., Revisiting text-to-image evaluation with gecko: On metrics, prompts, and human ratings, ICLR 2025

**Questions:**

- Can you clarify exactly how you implemented VQAScore? Your description in Section 2.3 ("decomposing prompts into questions") seems to contradict the standard implementation of that specific metric.

-  In Appendix A.11, you define three categories for aggregating human judgments. What percentage of your 12,832 pairs fell into the "High Disagreement" category requiring expert arbitration? Can you report standard metrics such as Krippendorff’s alpha reliability?

- How does TIT-Score distinguish between a model failing to generate a detail and the VLM simply failing to caption that detail? Are there ways to prevent VLM hallucinations from distorting the final score?

---

### Official Review · Reviewer_2fr7 · 2025-11-01

**Soundness:** 3
**Presentation:** 3
**Contribution:** 2
**Rating:** 4
**Confidence:** 4

**Summary:**

This paper introduces LPG-Bench, a benchmark with 200 long prompts (average >250 words) and 2,600 images from 13 SOTA T2I models with human rankings, to address the challenge of evaluating long-prompt image generation. The authors propose TIT (Text-to-Image-to-Text consistency), a zero-shot evaluation framework that converts generated images back to text via VLMs and measures semantic consistency with the original prompt. TIT-Score-LLM achieves superior alignment with human judgments, improving pairwise accuracy by 7.31% over the strongest baseline, while providing efficient and near-SOTA performance.

**Strengths:**

1. LPG-Bench is the first comprehensive benchmark for evaluating T2I model consistency with long prompts, comprising 200 curated prompts (average >250 words) and 2,600 images from 13 SOTA models with 12,832 human pairwise comparisons. It validates that long text understanding is a critical bottleneck for T2I systems and provides a solid evaluation foundation.

2. TIT (Text-to-Image-to-Text consistency) decouples cross-modal evaluation into two stages: (1) converting images to text descriptions via VLMs, and (2) computing semantic alignment between original prompts and descriptions. Offering two implementations—TIT-Score (embedding-based, efficient) and TIT-Score-LLM (LLM-based, accurate)—TIT achieves superior alignment with human judgments, with TIT-Score-LLM reaching 66.51% pairwise accuracy, a 7.31% absolute improvement over the strongest baseline, while providing near-SOTA performance with high efficiency.

**Weaknesses:**

1. While the proposed metric is reasonable, the fundamental motivation warrants scrutiny. Current user inputs are typically brief; when prompts are extended, LLMs typically handle expansion. In such scenarios, the LMM's expansion capability becomes the critical factor, not long-text comprehension per se. Does evaluating long-prompt consistency actually reflect real-world usage patterns and needs?
2. Although LPG-Bench is proposed, the work provides insufficient analysis of underlying causes. Why do certain models perform poorly on long prompts? The contribution amounts to benchmark construction and metric validation without deeper investigation or innovation. The paper lacks diagnostic insights into model failures, architectural limitations, or systematic factors affecting long-text understanding in T2I systems. A more compelling contribution would include root-cause analysis or design principles for improving long-prompt generation.

**Questions:**

What insights can LPG-Bench offer for subsequent model optimization? Why do we need to consider long text capabilities when user input is short text or text modified by LMM?

---

### Official Review · Reviewer_ozba · 2025-11-01

**Soundness:** 3
**Presentation:** 3
**Contribution:** 3
**Rating:** 6
**Confidence:** 4

**Summary:**

The paper introduces LPG-Bench, a benchmark aimed at evaluating text-to-image (T2I) models on long, detailed prompts. It comprises 200 prompts averaging >250 words, from which the authors generate 2600 images across 13 image generation models and collect human preference annotations. The study finds that common metrics (e.g., CLIP-score, LMM-score) correlate poorly with human judgments for long-prompt generation. To close this gap, the authors propose TIT (TIT-Score and TIT-Score-LLM) that compare the raw prompt to an LMM-generated description of the produced image. Experiments show TIT aligns better with human preferences.

**Strengths:**

- Good results: the experiment results show the alignment between TIT and human preferences largely outperform baselines.
 - Clear presentation: the plots (e.g. Fig 3), tables (e.g. Table 2) clearly explained the effectiveness of LPG-Bench and TiT.

**Weaknesses:**

- Dataset scale: 200 prompts may be small for a general-purpose benchmark.
 - Bias from LLM/LMM: TIT relies on an LMM to describe images, which might inject bias into the evaluation results.

**Questions:**

- Failure cases: it’ll be great if authors could qualitatively show some failure cases where TIT disagrees with humans, and discuss why TIT will fail on these use cases.

---

### Note · Authors · 2025-11-14

I have read and agree with the venue's withdrawal policy on behalf of myself and my co-authors.